# Effects of extreme melt events on ice flow and sea level rise of the Greenland Ice Sheet

Johanna Beckmann[1] and Ricarda Winkelmann[1,2]

[1]Potsdam Institute for Climate Impact Research, RD1, Potsdam, 14473, Germany,
[2]Institute of Physics and Astronomy, University of Potsdam, Potsdam, Germany

**Correspondence:** Johanna Beckmann (beckmann@pik-potsdam.de)

**Abstract.** Over the past decade, Greenland has experienced several extreme melt events, the most pronounced ones in the years 2010, 2012 and 2019. With progressing climate change, such extreme melt events can be expected to occur more frequently and potentially become more severe and persistent. So far, however, projections of ice loss and sea-level change from Greenland typically rely on scenarios which only take gradual changes in the climate into account. Using the Parallel Ice Sheet Model (PISM), we investigate the effect of extreme melt events on the overall mass balance of the Greenland Ice Sheet and the changes in ice flow, invoked by the altered surface topography. As a first constraint, this study estimates the overall effect of extreme melt events on the cumulative mass loss of the Greenland Ice Sheet. We find that the sea-level contribution from Greenland might increase by 2 to 45 cm (0.2 to 14 %) by the year 2300 if extreme events occur more frequently in the future under a RCP8.5 scenario, and the ice-sheet area might be reduced by an additional 6000 to 26000 $km^2$ by 2300 in comparison to future warming scenarios without extremes. We conclude that both changes in the frequency and intensity of extreme events need to be considered when projecting the future sea-level contribution from the Greenland Ice Sheet. Thereby it is crucial to resolve extremes individually on a high temporal resolution, as temperature forcing with the same excess temperature but evenly distributed over longer timescales lead to less sea level rise than for the simulations of the resolved extremes. Extremes lead to additional mass loss and thinning. This, in turn, reduces the driving stress and surface velocities, ultimately dampening the ice loss attributed to ice flow and discharge. Overall, we find that the surface elevation feedback largely amplifies melting for scenarios with and without extremes, with additional mass loss attributed to this feedback having the greatest impact on projected sea-level.

## 1 Introduction

With an ice volume of more than 7 metres sea-level equivalent, the Greenland Ice Sheet (GrIS) is one of the largest potential contributors to global mean sea-level rise (Morlighem et al., 2017). Since the 1980s, its ice loss has steadily increased with a mass loss increase of 80 Gt/year per decade (Mouginot et al., 2019), making up a significant share of the overall accelerating sea level rise (SLR) (Frederikse et al., 2020). These changes are driven by two main processes: (1) melting and accumulation that influence the surface mass balance (SMB) of the ice sheet and (2) changes in ice dynamics that influence the discharge into the ocean. Both processes have contributed almost equally to recent ice loss (Shepherd et al., 2020), with SMB changes dominating

since the year 2000 (Van Den Broeke et al., 2016; King et al., 2020; Slater et al., 2020). There are strong interactions between surface mass balance and ice dynamics, which jointly determine the total future SLR contribution of the GrIS. The decrease in SMB after the year 2000 has been attributed to enhanced melting in summer, which is related to a change of atmospheric circulation patterns over the North Atlantic (negative North Atlantic Oscillation index) (Delhasse et al., 2018). Whether the enhanced runoff was primarily driven by enhanced surface heating (Hofer et al., 2017) or reduced refreezing (Van Tricht et al.,

2016) is as of yet unclear.

Simultaneously, an increase of heat waves has been observed in the Northern high latitudes (Dobricic et al., 2020) while the number of extreme cold events declined (AMAP, 2021). This resulted in record breaking temperatures for Siberia in June 2020 (Overland and Wang, 2021) and May 2021 (Sullivan, 2021-11-10).Over Greenland, the cumulative heat wave index, representing the strongest heat wave in a year, has roughly tripled between 1981-1985 and 2011-2015 (Dobricic et al., 2020).

Observational data from 2000-2015 shows the probability of a heat wave occurring in the Arctic lies between 5-20 percent (Dobricic et al., 2020), meaning that heatwaves could be experienced roughly every 5 to 20 years in the Arctic.

In the past decade alone, Greenland has been subject to several extreme melt events, particularly in the years 2010 (Tedesco et al., 2011), 2012 (Nghiem et al., 2012) and most recently, during spring/summer 2019 (Tedesco and Fettweis, 2020). These extreme events greatly enhanced the surface mass loss (SMB) of the GrIS and are attributed to a strong negative North At-

lantic Oscillation index (NAO) in these summers, that led to persistent anticyclonic pressure heights over Greenland (so-called blocking events) (Hofer et al., 2017; Bevis et al., 2019; Tedesco and Fettweis, 2020).

In 2010, the GrIS experienced an early onset of the melt season with above normal temperatures, that led to a lowering of the surface albedo. Together with persistent warm temperatures and reduced snowfall, the long melt season led to the observed records of surface melt and runoff (Tedesco et al., 2011). Two years later, in 2012, a blocking pressure ridge over Greenland

was associated with the extreme event, that melted 98.6% of the ice surface on July 12th, including the summit. For comparison, on average 64.3% of the ice-sheet's surface area were melted between 1981-2010 (Tedesco and Fettweis, 2020). The year 2015 was another anomalous melt year, but the extent was quite different: While in 2012 almost the entire ice sheet was affected by the melt event, the event in 2015 was more localized but unusual since it was centered over the North of Greenland. This northward shift was again controlled by a stagnant pressure ridge in that area (Tedesco et al., 2016). Melting at the summit

historically happened every 150 to 600 years and the occurrence in 2012 thus could have been within the natural variability (Nghiem et al., 2012). However, in 2019, the summit melted again. Within three days 97% of the ice sheet surface was melted (Tedesco and Fettweis, 2020). Again, a persistent pressure ridge was associated with this melt event, but in contrast to 2012, it transited from Western Europe, where a heat wave was prevailing at the time (Cullather et al., 2020).

With progressing global warming, weather extremes such as these are generally expected to increase in frequency and intensity (Rahmstorf and Coumou, 2011). As of yet, however, it is unclear how such extreme events will affect the overall mass balance and future sea-level contribution from the Greenland Ice Sheet. Typically, the future sea-level contribution is assessed based on a gradual change in climate conditions using numerical 3D ice-sheet models. The models enable us to estimate changes in

surface mass balance and ice dynamics in response to different greenhouse gas emission and climate change scenarios. (The joint response of surface mass balance and ice dynamics is termed "full dynamics" throughout this manuscript.)

Based on the CMIP5 (Taylor et al., 2012) model ensemble for the Representative Concentration Pathways (van Vuuren et al., 2011), different studies assess a SLR contribution ranging from 1.5-4.9 cm (Goelzer et al., 2020) for the lowest emission scenario RCP2.6 at the end of the 21st century. For the highest scenario RCP8.5, the SLR contribution from Greenland within the 21st century is estimated at 7-21 cm (Church et al., 2013), 9-13.4 cm (Fürst et al., 2015), 4.6–13 cm (Calov et al., 2018), 14-33 cm (Aschwanden et al., 2019), and 4-14 cm (Goelzer et al., 2020), and for the year 2300, estimates range from 97 to 374 cm (Aschwanden et al., 2019). Although significant SMB changes are detected under global warming in the high-emission scenarios, none of the projections capture the recently observed changes of the negative NAO and the resulting extreme melt events on the Greenland Ice Sheet (Hanna et al., 2008; Delhasse et al., 2020). Moreover, observations show that the overall mass loss from Greenland at present is already higher than projected, which might at least in parts be due to the lack of considering extreme events in the simulations (Slater et al., 2020).

In a first study Delhasse et al. (2018) assesses the potential influence of an ongoing negative summer NAO under future warming. They simulated the observed circulation pattern from 2000 until 2017, repeatedly until until mid of this century, but with continuous warming at the boundary conditions. Thus they forced the atmosphere to the negative summer NAO and found a potential doubling of SMB loss compared to experiments with the same warming as boundary condition but no negative summer NAO. This approach estimates how such atmospheric conditions lead to generally warmer summers and include of course extreme summers as in e.g. 2012. However, this study does not disentangle the effect of the extremes alone and is limited to SMB changes only, neglecting the dynamic response of the ice sheet. The effect of future extreme melt events on the total SLR contribution (including changes in SMB and ice dynamics) of the GrIS therefore remains an open question. Here we assess this total contribution, including the changes in ice dynamics, for future extreme events of varying frequency and intensity. The ice dynamics considered in this study are focused on changes of ice flow due to changes in surface gradients or ice thickness. We do not consider changes in submarine melt rates or subglacial processes as e.g. glacial channel building that could in turn influence the basal sliding (Methods). As of yet, these processes are not well understood and require different experimental setups that would go beyond of the scope of this study (Discussion).

## 2 Methods

To project future mass changes of the GrIS, we use the polythermal and thermomechanically coupled Parallel Ice Sheet Model PISM (University of Alaska Fairbanks, 2019, 2019). We first introduce the model (Sect. 2.1) and describe its model initialization and calibration in Sect. 2.2. For our projection experiments we describe the derivation of temperature forcing scenarios with and without extremes in Sect. 2.3 and the calibration of the surface model to the temperature forcing. Finally, in Sect. 2.4, we present the different experimental settings that allow us to determine the dynamical response of the the GrIS to climate projections with and without extremes.

## 2.1 Ice Sheet Model PISM

The 3D high-resolution numerical ice-sheet/ice-shelf model includes a hybrid stress balance model (Bueler and Brown, 2009; Aschwanden et al., 2012) that combines the Shallow Ice Approximation (SIA) for vertical deformation (Hutter) and Shallow Shelf Approximation (SSA) for longitudinal stretching (Morland, 1987). The grounding line and calving front can generally move freely. In our experiments the ice sheet is expected to retreat. Thus, ice advance beyond the present-day ice margins is prohibited, by a strongly negative SMB (melting of ice) around the ice sheet mask to match the present-day ice sheet extent. This also applies for the calving front of glaciers, which are not allowed to advance. We also do not allow for floating ice thinner than 50 m at the calving front and use the von Mise calving law, appropriate for glaciers in Greenland (Morlighem et al., 2016).

In the applied model configuration, we concentrate on the effects of surface mass balance changes (SMB) and submarine melting is kept constant in time and space with a melt rate of 0.051914 m/year (PISM default setting). We do not consider changes in ice-ocean interaction via submarine melting as the resolution of most tidewater glaciers would require a finer resolution with a regional setting and we concentrate here on a first constraint for the Greenland wide mass changes. Furthermore, how extreme melt events would translate into ice-ocean interaction is not known and would be strongly modulated by the englacial and subglacial system of each glacier. Here, these englacial and subglacial processes as. e.g. the evolution of subglacial channels and their interactions with surface melt are not included. As of yet, many of these processes are not fully understood and their long term effect is unclear (Shannon et al., 2013; Tedstone et al., 2015). The SMB is calculated by a Positive Degree Day (PDD) model (Calov and Greve, 2005) with prescribed surface temperature and precipitation. PISM can simulate the surface elevation feedback by including a temperature correction for lowering surfaces (Zeitz et al., 2022). For this modeling option, the difference of surface elevation to the initial state is calculated and temperature is corrected with a lapse rate of $6°C$ per km. The corrected temperatures then modulate the SMB output via the PDD model accordingly. The projection experiments were run on a 4.5 km grid resolution, a vertical resolution of 20 m and monthly time steps. Bedrock and ice thickness data sets of BedMachine version 3 (Morlighem, 2019; Morlighem et al., 2017) were used to initialize and calibrate our model. Note that areas with ice thicknesses below 1m are excluded from our analysis when visualising our results. Bedrock deformation was not considered during this experiment.

## 2.2 Model Initialization and calibration

To obtain a realistic thermodynamic present-day state of the ice sheet, the temperature evolution around Greenland over the last glacial cycle was considered. The spin-up was therefore run over the last 125ka, with a scalar temperature anomaly (same temperature increase for every grid point on the ice sheet) of the 2D climatological mean field (precipitation and surface temperature) of 1971-1990, when the GrIS was close to balance (Mouginot et al., 2019). The historical temperature time series is derived from Oxygen Isotope Records from the Greenland Ice Core Project (GRIP) and can be found in the standard Present Day Greenland NetCDF files (Johnson et al., 2019). The climatological mean field was derived from MARv3.9 with ERA-40 (1971-1978) and ERA-Interim (1979-1990) of the ISMIP6 project (Fettweis, 2019a). Changes in precipitation were parame-

terized, with a 7.3% precipitation increase for each degree of surface warming, following previous approaches (Huybrechts,

2002). For computational efficiency we followed the grid refinement by Aschwanden et al. (2016): Starting in SIA-only mode, and a 18 000m grid at -125,000 years, we refined our grid to 9000m at -25,000 years and to 4500m at -5000 years. For the last -1000 years we keep the resolution fixed but added the SSA to the SIA stress regime, for better representation of the fast flowing outlet glaciers. The basal sliding velocities are related to basal shear stress via a pseudo-plastic power-law with a power of $q$ and the yield stress. The yield stress in turn follows the Mohr–Coulomb criterion, and is determined by models of

till material property (the till friction angle) and by the effective pressure on the saturated till. When switching to the SIA+SSA regime, we linearly altered the friction angle between $5°$ and $40°$ between -700m and 700m of bedrock elevation after Aschwanden et al. (2016). The resulting lower friction for lower altitudes and below sea level leads to an additional increase in surface velocities at the ice sheet margins, resulting in an improved match of flow structure for the glaciers. The state closest resembling present-day Greenland was achieved for the following ice flow parameters: flow enhancement factor E = 3 (for the

entire spin-up), exponent of the sliding law q = 0.6 and exponent of the flow law for the SSA n = 3 (for the last 1000 years). All other parameters we set to default (University of Alaska Fairbanks, 2019, 2019).

The initial state as modelled here has a total ice volume of 7.6 m sea-level equivalent (Fig. 1a). On average, ice thicknesses deviate by 170 m with a root mean square error of 238 m to the observed state. In our initial state the ice thickness is overestimated along the margins and in the south-west and north-east of Greenland while the north-east, ice interior and south-east

are underestimated (Fig. S1a). Strongest relative misfits occurs at the margins of Greenland (Fig. S1b). Complete observational velocity data for the entire ice sheet is not available for the time period 1971 to 1990. We therefore compare with the complete velocity data set by Joughin et al. (2018) that gives the average velocities from 1995 to 2015. Our comparison (Fig. S2a,b) shows an overall agreement of the velocity pattern with an average difference between modelled and observed ice speed of 9 metres per year and a root mean square error of 146 metres per year. Stronger deviations occur mainly in the fast flowing

glacier regions (Fig. S2c) where the exact position of the glaciers and its catchment area is crucial and a coarse resolution leads to a poorer agreement with observations (Aschwanden et al., 2016). This leads to a RMSE of 413 m/yr for regions flowing faster than 100m/yr and a strong relative error at the margins of the GrIS (Fig. S2d). After reaching the stable spin-up state, our projection experiments were run from the year 1971 onwards.

## 2.3   Temperature forcing

Starting from an initial state of the Greenland Ice Sheet under present-day boundary conditions (Fig. 1a), we apply surface temperature trajectories based on reanalysis data (Sect. 2.3.1) and the RCP8.5 emission scenario (Sect. 2.3.2). We here choose the RCP8.5 scenario since latest results from satellite observations and regional climate models show that the speed with which Greenland is currently losing ice would only be expected under this highest emissions scenarios (Slater et al., 2020).These is our baseline scenario and we further add extremes of different frequency and intensity (Sect. 2.3.3). After the derivation of the

forcing scenarios we calibrate our PDD model to the observed SMB loss (Sect. 2.3.4).

### 2.3.1 Forcing until 2018

To project the mass changes of the GrIS with PISM, we applied changes in a scalar temperature field derived for the average surface temperature anomaly over Greenland with MARv3.9 (Fettweis et al., 2013, 2017) from ERA (1971-2017) to our spin-up state. We used the ERA Interim data set as it is closest to the observation and includes already some observed extremes. A comparison to observational mass changes (Mouginot et al., 2019) can be found in Fig. S3, showing that our fully dynamic simulations of the ice sheet (red line, Fig. S3) agree reasonably well with the observed mass balance of the GrIS between 1972-2017 (red dashed line, Fig. S3), however not capturing the strong slope from 2000-2017. Figure S3 also shows the observed surface mass balance changes (Fig. S3, blue dashed line), which are underestimated by our PDD model (Fig. S3, blue line). However, the estimates SMB loss from Mouginot et al. (2019) also do not agree well with the original (1km resolution) MAR SMB changes (Fig. S3 ,orange line) as Mouginot et al. (2019) use a different regional climate model to estimate the SMB changes. We here concentrated on tuning the PDD model to fit the SMB changes of the entire century of MAR output, as tuning it to the historic changes would drastically underestimate SMB loss in the future (Sec. 2.3.4).

### 2.3.2 Temperature Forcing without extremes

For the temperature evolution without extremes, we use the Greenland-wide averaged surface temperature anomalies as projected by MAR (Fettweis et al., 2013, 2017) based on the MIROC5 RCP8.5 scenario (Fettweis, 2019b) from 2018 until 2100, and extended to the year 2300 ( see below, Fig. 2a), called MIROC5 hereafter. Monthly temperature anomalies averaged over Greenland may increase by up to 10 °C by the end of this century (Fig. 2) based on the MIROC5 results. Note that this corresponds to a global mean temperature change of 4 deg by 2100 (Fig. S4a); the warming signal over Greenland is significantly stronger due to polar amplification. In July, temperature increases of about 15 °C (Fig. 2) are reached within 2300.

We extended the different scenarios for the 21st century described above until the year 2300: As the MIROC5 output was only available until the year 2100, we derived the annual temperature anomaly for the GrIS until 2300 by interpolating from the emulated annual global mean temperatures (GMT) of MIROC5 by Palmer et al. (2018) (Fig. S4). To this end, from the annual MIROC5 results until 2100 we first derived a quadratic trend function ($T_{GrIS,trend} = 1280.16°C - 1.31°C/year \cdot years + 20°C/year^{-2} \cdot years^2$) to exclude the inter-annual variability (Fig. S4a). Together with the GMT until 2100 we determined a fitting function (Fig. S4b) $T_{GrIS,emulated\ trend} = 0.1°C + 0.96°C^{-1} \cdot GMT + 0.15°C^{-2} \cdot GMT^2$ ,in order to emulate $T_{GrIS,trend}$ beyond the year 2100 in dependence of the GMT. Thus, with the GMT until 2300 and the fitting function, we established the GrIS trend function until 2300 ($T_{GrIS,emulated\ trend}$, red dashed line). To this, we added the inter- and intra- annual variability to receive a more realistic monthly temperature projection. This was done by first calculating the yearly anomalies of the 2050-2100 from the fitting function to the actual annual values: $\Delta T_{yr(2050-2100)} = T_{GrIS(2050-2100)} - T_{GrIS(2050-2100),trend}$ and then randomly picking out of $\Delta T_{yr(2050-2100)}$ and adding on to the emulated trend $T_{GrIS,emulated\ trend}$ until the year 2300. This gave us the new annual temperature curve until the year 2300 with inter-annual variability ($T_{GrIS,emulated}$ red solid line, Fig.S 4a ). However, as we need monthly tempera-

tures we recalculated the monthly temperature values for our newly created $T_{GrIS,emulated}$ by adding the monthly temper-
ature anomalies as well. Thus, for each annual anomaly $\Delta T_{yr(x)}$ we picked, we calculated its monthly anomalies by col-
lecting the full 12 months of that year and subtracting them by the mean of the annual trend (and not the annual mean)
($\Delta T_{x(1,...,12)} = T_{GrIS(x(1,..,12))} - T_{GrIS(x),trend}$). Thus each annual anomaly contains now monthly anomalies as well
$\Delta T_x = \Delta T_{x(1,...,12)}$ that were added to the emulated trend. These values served as our baseline scenario until the year 2300
(Fig. 2, dark grey lines).

The change from a spatially explicit to a uniform temperature forcing adds biases, especially at the margins of the ice sheet.
Figure S5 shows an overestimation of mass loss at the Western margins in the year 2100 when using uniform forcing. How-
ever, in both simulations the total ice volume is almost equal, with the ice volume from the simulations of the spatially explicit
temperature anomalies being only 0.4 % larger than the scalar temperature anomaly field, indicating a similar SLR in 2100
for both cases. Over time, SMB losses of these two approaches differ quite as well, but for our ice sheet wide estimation, the
SMB loss calculated by the PDD model agrees better with the MAR SMB loss when using such a scalar temperature field (SI,
Fig. S12).

### 2.3.3 Forcing scenarios with extremes

The MIROC5 temperature trajectory serves as a baseline for the development of the scenarios including temperature extremes.
As it is highly uncertain when and at which location exactly extremes are most likely to occur under progressing climate
change (Otto, 2016, 2019), we here develop idealized extreme scenarios of different intensities and frequencies only altering
temperatures in July: The extremes are added from July 2012 onwards, occurring every 20 (scenario $f_{20}$), 10 (scenario $f_{10}$),
or 5 (scenario $f_5$) years, based on the observed heat wave probabilities of 5 to 20 % in the Arctic at present (Dobricic et al.,
2020). While in reality extremes occur irregularly, we choose these regular intervals to be able to extract the effect of extreme
frequency systematically. The different intensities of our extreme event scenarios are reached by multiplying the 10 year
average of MIROC5 temperature anomalies in July by factors of 1.25 (scenario $I_{1.25}$), 1.5 (scenario $I_{1.5}$) and 2 (scenario
$I_2$). We choose to use a multiplier here instead of adding a constant temperature offset, since the magnitude and variability
might increase in the future and, e.g., the 2012 extreme event will eventually become the "new normal" under continued global
warming in the future. Furthermore, this approach allows us to assess the influence of different intensities in a systematic
manner and is easily reproducible for further studies. Section 2.1 in the SI explains in detail how our intensity factors compare
to a potential changing climate in the future.
With our running mean approach, the factor 1.5 ($I_{1.5}$, Fig. 2) leads to a temperature increase comparable to that of the extreme
melt year 2012 ($\Delta T_{ERA}(2012) = 1.8\,°C$, $\Delta T_{I_{1.5}}(2012) = 1.7\,°C$). We therefore use it as the default scenario in this study.
Thus $I_{1.25}$ (Fig. S6) and $I_2$ (Fig. S7) give peak temperatures that lie below and above the observed one in 2012, respectively.
In the default scenario, temperatures in July can reach changes of above 20 °C at the ice-sheet surface ($I_{1.5}$, see Fig. 2b).
The lower-intensity ($I_{1.25}$) and the higher-intensity ($I_2$) scenarios lead to temperature increases of around 15 °C and 30 °C

respectively (Fig. S6 and Fig. S7). To ensure comparable future SMB calculation between our PDD approach and MAR, we calibrated the PDD model within PISM accordingly.

### 2.3.4 PDD calibration for future temperature forcing

Since our projections are forced with a scalar temperature field, precipitation changes are determined based on the present-day pattern, modifying it by a precipitation rate parameter. This approach allows different magnitudes of precipitation but does not consider changes in the precipitation patterns themselves. We derived a precipitation rate of 5% per degree of warming, from the weighted monthly mean of the ERA Interim data set for 1971-2018 (Fig. S10).

The PDD model determines the runoff by calculating the melt from snow and ice and the amount which is not refrozen with 230 the refreezing parameter. The default setting leads to a substantial underestimation of the future melt compared to the MAR-MIROC5 output. To ensure comparability of future melt changes between the PDD model and the MAR output, we derived a temperature-dependent refreezing parameter from the MAR-MIROC5 daily output. (Fig. S11).

With the determined refreezing parameter, we calibrated our PDD model with the monthly 2D temperature and precipitation fields from MIROC5 until 2100. The closest fit to the MIROC5 SLR-equivalent via SMB loss was achieved with a temperature 235 standard deviation of $\sigma_T = 5\,^\circ\mathrm{C}$ (Fig. S12). We use the default PDD melt factors for snow and ice of 0.00329 and 0.00879 meters water equivalent per degree per day, respectively. SLR was calculated by the cumulative sum of the SMB anomalies to the 1971-1999 SMB mean.

Thus, all future experiments were run with a precipitation increase of 5% per degree of warming which is in line with other studies, a temperature standard deviation of 5 $^\circ$C (equals PISMs default value) and a temperature dependent refreezing function 240 of $23.42\% - 1.34\frac{\%}{^\circ C} \cdot \Delta T$ (PISM's constant default values is 60%).

### 2.4 Experiments for the dynamic response

We derived a set of 10 different temperature forcing scenarios, that include a MIROC5 baseline scenario and its 9 versions of extremes. These extremes differ by three intensities $(I_{1.25}, I_{1.5}, I_2)$ with each having 3 different frequencies (5, 10 and 20 years). To quantify the dynamic response of the GrIS we derive, for each of the future temperature scenarios three different 245 kinds of SLR-projections:

- (1) Full dynamics: To estimate the total mass loss (SMB and ice dynamics) PISM was run with the temperature scenario and a surface temperature lapse rate of 6 $^\circ$C per km which lies in the range of GrIS-wide average lapse rates of the MAR output (Fig. S13).

- (2) SMB-only: To estimate the impacts of SMB changes without any dynamic response from the ice sheet, PISM was 250 run similarly but no dynamic changes were allowed and the ice thickness was held constant. Thus SMB changes were calculated with PISM's internal PDD model for a constant surface topography but with changing temperatures. These SMB changes were converted to mass loss for ice thicknesses above flotation, which in turn gave the projected SLR.

- – (3) Dynamics without surface elevation feedback: To approximate the role of the ice sheet dynamics excluding the melt-elevation feedback, PISM was run as in the full dynamics case, but without including the surface temperature lapse rate correction.

For each SLR projection, the control run conditions was subtracted. The control was run under the full dynamic experiment with fixed boundary conditions (i.e. no temperature anomalies).The sea-level rise projections for the simulations including ice dynamics was calculated from the PISM variable 'sea-level potential', which gives the potential global sea-level increase if all ice above flotation was melted and distributed as freshwater over the ocean area estimated at $362.5 \ 10^6 \text{km}^2$. For the SMB-only runs we added the SMB differences from the SMB of the climatological mean 1971-1999 (from our spin-up state) for each grid cell and subtracted the corresponding ice mass change from the thickness above flotation. The calculated ice loss was then converted to sea-level rise equivalent as for the dynamic runs. Since our dynamic control run shows a slight decrease in ice volume (SLR equivalent of 5 cm in 2100 and 10cm in 2300 Fig. S14), our spin-up state is not in perfect balance and we therefore subtracted the drift from all simulations.

## 3   Results

### 3.1   Additional sea-level rise due to extremes

Based on the temperature scenarios described above, we use the fully-dynamic Parallel Ice Sheet Model PISM (see Methods) to assess the impacts of extreme events on the future sea-level contribution from the Greenland Ice Sheet. This includes changes in the surface-mass balance itself, accelerated flow through the steepening of surface gradients and their interactive feedbacks. We find that under all scenarios, the extreme events generally lead to an increase in ice loss in our model simulations with full dynamics. Compared to the climate change scenario without extremes, the 3.08 m SLR by 2300 determined from MIROC5 is increased to 3.13 m ($f_{20}$), 3.18 m ($f_{10}$) and 3.28 m ($f_5$), or by 1.6 to 6.5 %, for the $I_{1.5}$ scenarios (see Fig. 3a, and Table 3). The lower intensity $I_{1.25}$ scenarios only lead to slightly higher SLR contributions of 3.10 m ($f_{20}$), 3.12 m ($f_{10}$) and 3.17 m ($f_5$) (see Fig. 3b) which is substantially increased for the higher intensity $I_2$ scenarios with 3.19 m ($f_{20}$), 3.30 m ($f_{10}$) and 3.52 m ($f_5$), respectively (see Fig. 3c). Thus the most severe scenario with extreme events of higher intensity occurring every 5 years ($I_2$, $f_5$) results in an increase of projected SLR by almost half a meter compared to the scenario not including extreme events.

The projected sea-level contribution from Greenland is generally much lower within the 21st century, where the climate change scenario without extremes leads to ice loss equivalent to 0.13 m sea-level rise (Table 1). At this order of magnitude, the additional mass loss due to extreme events is not as pronounced as in the year 2300. Only for the higher intensity $I_2$, we find additional SLR of 1 and 2 cm for frequencies of 10 and 5 years, respectively.

For the same intensity, we further test how long after a particular extreme event significant effects can still be detected in the ice-sheet mass balance. To this end, we apply the forcing including extreme events up until the year 2100 but not thereafter and then compare the respective sea-level responses. Our results suggest that it takes about 20-30 years until the SLR trajectory is closer to its original track (i.e., closer to the MIROC5 scenario) than to the extreme event scenario, and more than 100 years until the effect from the extreme events is diminished to less than 10 % (Fig. 4). With the preceding $I_2, f_5$ extreme scenario until 2100, the final SLR by the year 2300 without continued extremes is 3.09 m, almost equal to the MIROC5 scenario (3.08m).

## 3.2 Importance of ice dynamics

In order to quantify the relative importance of ice-sheet dynamics, we compare the full-dynamic simulations conducted with PISM to SMB-only simulations, keeping all other parameters fixed (Sect. 2.4). Including ice-dynamic effects generally increases the projected ice loss from Greenland ( Fig. S14 and S15, see also Tables 1 and 3), as the dynamical ice flow transports ice from the interior towards the margins of the ice-sheet, thus delivering more ice to the ablation zone where it can subsequently melt. Furthermore, enhanced surface melt can eventually onset the melt-elevation feedback, where the lowered surface elevation exposes the ice to warmer temperatures, in turn leading to more melting. Based on the MIROC5 results, the SMB-only scenario leads to SLR of 0.08 m by 2100, which increases to 0.13 m when including ice dynamics. Neglecting the melt-elevation-feedback would result in a total of only 0.11 m SLR, i.e., 2 cm less than when considering the full dynamics (see Table 1). For the $I_2, f_5$-SMB only scenario we experience an average yearly increase of SMB loss of a factor 1.2 (Table 2) compared to the baseline SMB-only scenario in the time frames 2012-2037 and 2038-2048 when average summer temperature (in the base line scenario) is increased by 1.5 and 2 °C respectively (Fig. S16).

The amplification of the sea-level response due to ice dynamics is pronounced even more in the extended projections until the year 2300: For our MIROC5 simulations, the SMB-only scenario leads to SLR of 2.27 m, while the full-dynamic response of the GrIS yields 3.08 m SLR in our simulations, translating into a 35 % increase compared to the SMB-only scenario (Table 3). The same dynamic enhancement occurs for the extreme event scenarios, where the full-dynamic simulations result in higher sea-level projections compared to the SMB-only case (see Tables 1 and 3).

The dynamic experiments that do not consider the surface elevation feedback lie in between our SLR projections for the SMB-only and full dynamic experiment but are only slightly higher than the SMB-only runs (about 1 % higher). For this constellation, the dynamic mass loss adds only a little more SLR than the pure SMB loss. This clearly shows the importance of the surface elevation feedback, i.e. it is important to include the effects of both the surface-mass balance and the ice dynamics in sea-level projections in an interactive manner.

Overall, the effect of including ice dynamics with is greater than the increase between the different extreme event scenarios considered here (Fig. S14 and S15 ), mainly due to the impact of the surface-elevation-feedback. Further, it underlines how crucial it is to assess the full dynamic response of the GrIS, including impacts with respect to ice retreat and changes in ice velocity.

### 3.3 Ice-margin retreat and velocity changes

Ice loss primarily happens at the margins of the ice sheet, were melting is more prominent and temperatures are higher due to the lower surface elevations than in the ice-sheet interior. As the topography is lower in the west than the east of Greenland, the west is generally also more exposed to higher surface temperatures and more vulnerable to the melt-elevation feedback. Consequently, we find the strongest retreat and decline in ice area in Western Greenland as well (Fig. 1b). To describe the differentiated behaviour of the ice sheet for the fully dynamic run, we introduce different basins (Rignot and Mouginot, 2012; IMBIE2016, 2019): North(NO), North-East(NE), South-East(SE), South-West(SW),Central-West(CW) and North-West(NW) as depicted in Figure (1). The South-West region shows the furthest retreat (Fig. S17,Fig. S18) with a total ice loss of $258 \ 10^3\text{Gt}$ by 2300 (Fig. S18). Approximately the same amount is lost in the North-East region (Fig. S18), but constitutes only a smaller fraction of its present-day ice volume (Fig. S19).

In total, the Greenland Ice Sheet loses about $652 \ 10^3\text{km}^2$ of ice area by 2300 in the scenario without extreme events (Table S1), that is roughly 36 % of its area at present (control run: $1795.12 \ 10^3\text{km}^2$, see Fig. 1a).This area is further reduced by 6 and $14 \ 10^3\text{km}^2$ for extremes occurring every 20 and 10 years (for the default $I_{1.5}$ scenario, see Table S2). For increased frequencies, with extreme events occurring every 5 years ($I_{1.5}, f_5$) the ice-sheet area is reduced by an additional $26 \ 10^3\text{km}^2$, resulting in a more pronounced ice retreat at the margins of the Greenland Ice Sheet (Fig. 1b). The lower intensity $I_{1.25}$ scenarios lead to less additional area losses of roughly half the ones detected for $I_{1.5}$, while extremes with the highest intensities $I_2$ approximately double the additional area loss.

At the same time, the reduced ice area in 2300 is accompanied by an acceleration of ice velocities from an average $25 \ \text{m/year}$ to $52 \ \text{m/year}$ (for the $I_{1.5}, f_5$ case, see Fig. 1d). Areas where the ice velocities are faster than $100\text{m/year}$ comprise roughly $18\%$ of the Greenland Ice Sheet by 2300 in our simulations. This corresponds to an increase by a factor of 1.5 compared to the initial state (where roughly $12\%$ of the ice flows this fast (Fig. S20). However, in the year 2300, glaciers with maximum velocities higher than 500m per year are lost due to the overall ice retreat (Fig. S20,S21). Changes of the ice sheet differ for specific regions. The average ice sheet wide speedup is therefore mainly caused by a moderate speedup of the ice area around the catchment areas of the glaciers (Fig. 1d) and the continuous speedup of the CW sector (Fig. S22). The CW-sector is the only sector in which average surface velocities continue to increase until 2300 (Fig. 6,Fig. S23,yellow line) with a large part of the glaciers remaining in contact to the ocean, keeping a reduced basal friction and therefore high basal velocities (Fig S24) while other sectors become land terminating. Acceleration can be attributed to an increased gravitational driving stress ($\tau_d$), which is a function of the ice thickness($H$) and surface gradients ($h_s$): $\tau_d \propto H h_s$. The speedup of the CW-sector throughout the warming scenario is driven by a strong steepening in surface slope that counteracts the overall thinning, thus increasing the driving stress (Fig. 5). The SW sector also experiences a steepening of its ice surface. The average thinning however, diminishes this effect on the driving stress and thus surface velocities increase only moderately until 2250. After 2250 very little ice remains in the SW-sector, leaving only slow-flowing ice close to the summit. Surface steepening also leads to a small speedup in the NE until 2300 and the NO until 2200, where thinning then results in a decrease in driving stress and subsequent

slowdown. The NW and the SE sector both experience a deceleration of surface velocities until 2300 due to a decrease in driving stress. While this is due to a decrease in surface slope and even thickening in the SE-sector until around 2200, the NW sector steadily thins, leading to the decrease of driving stress. All sectors experience a thinning. However, due to the different evolution of velocity not all sectors show a decrease in the mean ice flux (Q, Fig. S23), which can be approximated by the mean thickness multiplied by the mean vertically integrated velocity. The CW shows a strong increase in ice flux towards 2300. Only little changes can be observed in the NW and NE sector while reduction in Q is observed for NO,SW and SE. Overall we find lower average sector velocities with increasing intensity of extremes (Fig. 6) due to the additional thinning and reduced driving stress induced by the enhanced SMB loss.

The role of the surface elevation feedback for these simulations can be estimated when comparing the full dynamics runs with the dynamic runs without surface elevation feedback in Figure S25 for the baseline scenario. In the year 2100, thinning is more prominent along the margins and the southwest of the GrIS for the full dynamic runs. The thinned ice cells decrease in surface velocities, but further inland surface velocities speed up due to the steepening gradient. This effect is even further amplified in 2300: Thinning is more and more amplified, reaching into the ice interior of the GrIS leading to further retreat. Steepening the interior leads to speedup while thinning at the margins reduces surface velocities. Dynamic runs without the surface elevation loose only about a quarter of the present day ice sheet area ( $440.3 \ 10^3 \mathrm{km}^2$ Tab S3). Additional retreat due to extremes is about $4.5, 9$ and $18 \cdot 10^3 \mathrm{km}^2$ for $I_{1.5}$ with frequencies of 20,10 and 5 years respectively (Tab S4). Here as well, we see roughly a doubling and halving of the additional mass loss for the other intensities ( $I_2$ and $I_{1.25}$ respectively).

Likewise, the surface elevation feedback shows a bigger effect on ice sheet retreat than the additional extremes.

### 3.4 Role of intensity and frequency of extreme events

Our results show that extreme events can add considerable mass loss to existing projections from Greenland. Herein, both the intensity and frequency of the extreme events play a crucial role in determining the future evolution of the ice sheet (Fig. 7): In our model simulations with full dynamics, a doubling in frequency leads to twice as much additional SLR for all intensities. At the same time, increasing the intensity from 1.25 to 2, enhances the additional ice loss by roughly a factor of 5. In combination, a higher frequency and intensity have an even stronger effect than their sum.

Considering solely the changes in SMB, we find a similar relationship, with relative ice loss doubling for a doubling in frequency, and increasing by a factor of 5 for a change in intensity from 1.25 to 2 (Fig. 8). Thus the relative changes of the SMB-only and the full dynamic run are approximately the same (Fig. S26 and S27). The dynamic run without the surface-elevation feedback adds an additional 3-4cm in 2300 to the SMB-only scenario regardless of the intensity of the extreme (see Table 3).Thus, for the dynamic run without surface elevation feedback, SLR increase is the same as for the SMB-only scenario (Fig. S28) but the relative increase is smaller compared to the SMB-only scenario (Fig. S29). Independent of each PISM experiment, the increase in frequency and intensity leads to a further doubling of additional SLR. The magnitude of the additional SLR, however, is strongly amplified by the surface elevation feedback and gives overall higher absolute values for

all temperature scenarios. However, independent of this feedback the impact of extremes play a similar role for all experiments (SMB-only,dynamic and full dynamic).

## 3.5 Resolving extremes

The question arises whether the additional temperature excess inserted by the extreme is important to resolve on this monthly time step or whether the evenly distribution of this excess temperature would lead to the same impact in terms of SLR. In other words, do extremes really matter? We therefore averaged the original extreme temperature forcing of a monthly resolution over different time frames and assess their contribution to SLR (Tab. 4).

The averaged temperatures were recalculated to their monthly equivalent to produce a smoothed data set but with a monthly output. This allows us to keep the same experimental setting of a monthly time step for PISM as in our original experiment and exclude any numerical biases that might be introduced by e.g. changing the time step of PISM. We concentrated on investigating the most severe scenario ($I_2, f_5$).

Averaging the temperature anomalies over one year led to reduced SLR in 2300 of only 3.12m compared to the 3.52m SLR of the original experiment. To disentangle the effect of the extremes and the effect of the averaging, the yearly mean of the baseline scenario without extremes was run as well. SLR resulted in only 3.01m compared the 3.08 of the original, and thus the additional SLR of the extremes for the yearly means is only 11cm compared to the 44cm of the monthly means. This is unsurprising, as the temperature increase is evenly distributed over the year and not concentrated within the melt season. We therefore ran another set of experiments where the temperature was averaged over the summer (June, July and August).

For the baseline scenario this leads to 3.07m SLR. Compared to the 3.08m SLR of the original MIROC5 experiment, this is only a minor difference and could lead to the assumption that the mean summer temperature is enough to project SLR accurately. However, for the extreme scenario ($I_2, f_5$), using a summer mean gives a SLR of 3.43, 9cm less than the original experiment. The additional SLR due to extremes is even further reduced (to 32cm) when averaging the access temperature over 10 summers.

Our findings suggest that a monthly resolution is important to quantify the effect of extremes. However, this effect only comes into play in the year 2300. For the lower temperatures in the year 2100, SLR differs only by 1cm for the various experiments, even for the yearly averages.

## 4 Discussion

We examined the impact of extreme events on future sea-level rise due to the Greenland Ice Sheet, using the Parallel Ice Sheet Model PISM. Overall, we find that along with progressing climate change, the more frequent occurrence of extreme events plays a crucial role in determining the future sea-level contribution of the Greenland Ice Sheet. Our results suggest that over time, extreme events can lead to an additional retreat of the ice-sheet margins and additional ice volume loss compared to the baseline climate change scenario without extremes. Taking severe extreme events ($I_2, f_5$) into account can increase the projected sea-level rise by up to half a meter (14 %) by the year 2300 compared to the MIROC5 scenario without additional extremes.

Thereby, both the intensity and frequency of the extremes play an equally important role. Furthermore, we show that resolving the extremes on adequate model time steps has to be considered, as simulations using coarser time steps albeit reaching the same seasonal average temperature lead to less projected SLR. Compared to the baseline scenario, surface velocities in turn can decrease when including more intense extreme events, due to the reduced driving stress invoked by the additional SMB loss and thinning.

We also investigated the importance of ice dynamics, by comparing full-dynamic simulations with SMB-only simulations, and dynamic simulations without the surface elevation feedback. We found that the effects of extremes on additional SLR was similar across all kinds of different experiments (SMB-only, full dynamic and dynamic without surface elevation feedback): A doubling of intensity or frequency led to roughly a doubling of additional SLR. However, including ice-dynamic effects generally increases the projected ice loss from Greenland, as the dynamical ice flow transports ice from the interior towards the margins of the ice-sheet, thus delivering more ice to the ablation zone where it can subsequently melt. A lowering of the surface due to melting can onset the surface elevation feedback, leading to enhanced melting. Hence, mass loss is greatest for the fully dynamic runs and amplifies SLR by over 30% , when comparing it to the SMB-only scenarios in 2300 while dynamic runs without the surface elevation feedback only add a bit more than 1 %. Thus, when approximating SLR in 2300 simulation for our baseline scenario, the SMB only experiment would give 2.27m SLR but including the surface-elevation-feedback would add roughly 1 meter additional SLR and including the most server extreme scenario would add another half meter of SLR.

### 4.1    Comparison to other studies

The importance of considering extremes in future projections of the Greenland Ice Sheet is further supported by other studies (Mikkelsen et al., 2018), that show a strong effect of temperature variability on the GrIS equilibrium volume. The only other study that truly captures extremes as well is the one from Delhasse et al. (2018). As they use a regional atmospheric model we compare our results to the SMB-only runs. Compared to them, our experiments conducted here are rather conservative (at least on the short term) in the sense that the simulated SMB loss only increases by a factor of 1.2 and not 2 as by Delhasse et al. (2018) for comparable average warming. However, this can be attributed to the fact that Delhasse et al. (2018) approach re-simulated the total summer variability of 2000-2017, while we here only increase the variability in July by extremes. Furthermore, it demonstrates that the projected MIROC5 variability is in general lower in the beginning of the century than the ERA observations from 2000-2017. Figure S16 demonstrates the substantial increase of 0.7-0.8 °C in summer mean temperatures (on top of the average 1.5 °C and 2 °C), if we were to add the variability of the entire summers in 2000-2017 as from ERA observations (Dee et al., 2011). Adding the entire summer variability from ERA observations would clearly lead to more SMB loss and bring our results closer to the one of Delhasse et al. (2018). However, this method would not investigate the effect of extremes alone nor would it consider an increase in their variability throughout the century.

To compare with other SLR projections, we only look at our fully dynamic runs.: For the year 2100, our projected SLR without extremes of 13cm lies in the upper range of other estimates (Fürst et al., 2015; Calov et al., 2018; Goelzer et al., 2020) and well within the estimates in the 6th Assessment Report of the IPCC(Fox-Kemper and Yu, 2021) of 9-18cm for SSP5-8.5. Only our

simulations with extremes match the lower range of Aschwanden et al. (2019) (14-33cm) for the year 2100, who used a PISM -PDD approach as well but considered ocean warming and submarine melting. In the same study the optimal simulation, that best reproduced the 2000–2015 mean surface mass balance, led to an estimated SLR of 174 cm in 2300 with an uncertainty range of 97-374 cm(Aschwanden et al., 2019). Our simulations without (308cm) and with extremes (352cm) lie thus in their upper uncertainty range.Overall our simulations without extremes mostly lie in the upper range of other SLR estimates. The additional SLR increase introduced by extremes, results in a moderate uncertainty compared to the study of Aschwanden et al. (2019).

## 4.2 Limitations

In this study, we looked at Greenland-wide heat events with a spatially uniform temperature increase, potentially adding a bias of higher mass loss at the western margins of the ice sheet (until the year 2100, where 2D data-ouput was available for comparison) compared to a use of the spatially distributed temperature changes. Future work could concentrate on more local melt events and the vulnerability of specific regions in Greenland. This spatially uniform temperature increase also leads to a bias in the initial state compared to present day. Difference in thicknesses and coarse spatial resolution lead to higher root mean square errors in the velocity pattern as compared to e.g. Aschwanden et al. (2019, 2016). However, there the authors correct their initial state by applying a correction flux that additional decreases or increases ice mass to match the one in present day and very fine resolution can improves the simulated velocities. Our simulated biases might influence the regional mass loss as ice simulated thicker than observation might melt later in time and changes in the driving could then in turn influence ice sheet flow differently. For a more regional investigation, future studies could include such a flux correction and study extremes on a higher spatial resolution but the effect on the tuning of the PDD model would have to be investigated as well. However, we think that the overall ice sheet wide effect of extremes would lead to similar results: An increase in in mass loss due to surface mass loss and the surface elevation feedback and a decrease in surface velocities due to a reduction in the driving stress.

Our simulations do not include effects from ice-ocean interaction which may play a crucial role with respect to the dynamic changes of glaciers at the margins of the GrIS (Beckmann et al., 2019; King et al., 2020). The increased surface runoff during the next centuries can clearly influence the subglacial hydrology via the formation of subglacial channels which in turn can influence basal sliding. Our experiments due not consider any changes of this nature, as many of these processes are not fully understood and their long term effect is still unclear (Shannon et al., 2013; Tedstone et al., 2015).

Further, we here use a PDD model to simulate changes in the surface mass balance, which is only temperature-dependent and does not consider changes in humidity, radiation, pressure or albedo like the more complex energy balance models (Krebs-Kanzow et al., 2018) or regional climate models (Fettweis et al., 2017; Noël et al., 2018). Including the response of albedo changes might show significant effects of mass loss in response to different timing of the extreme event as e.g. the beginning or ending of the melt season. Likewise, potential changes in precipitation patterns were neglected and only temperature-dependent changes in precipitation (based on the pattern from 1971-1990) included. Our experimental setup generally shows an enhanced sensitivity towards the surface elevation feedback, most probably because of using a temperature dependent refreezing param-

eter. For the years 2100 and 2300 we detect a 18% and 33% increase of SLR from the dynamic simulations without and with
the surface elevation feedback. Other experiments using one-way coupling of the regional climate model MAR and an ice sheet
model, where the SMB changes are calculated with MAR and the surface-elevation feedback is considered, show an increase
of 4% in 2100 (Edwards et al., 2014), 8% in 2150 (Le Clec'h et al., 2019) and approximately 10% in 2200 (Edwards et al.,
2014; Delhasse et al., 2023) but for a stable warm climate after 2100. Despite these limitations however, the PDD model pa-
rameters were tuned to best match the ice sheet-wide SMB evolution calculated by regional climate model MAR, and our SLR
projections based on the scenario without extremes lie within the same order of magnitude compared to former projections
until year 2100 (Fürst et al., 2015; Calov et al., 2018; Goelzer et al., 2020) and 2300 (Aschwanden et al., 2019).

## 5 Conclusions

Previous studies did not consider extreme melt events when projecting sea-level rise (SLR), and predictions of weather
and climate extremes are generally accompanied by high uncertainty (Otto, 2016, 2019). Hence, our idealized experiments
offer an initial assessment of how future extreme events may impact the Greenland Ice Sheet, highlighting the significance
of incorporating extremes in future SLR projections. It is essential to take into account both the intensity and frequency of
extreme events in future projections. Comparatively, our most severe scenario, in contrast to a scenario without additional
extremes, could result in an additional SLR of approximately half a meter or 14% within a fully dynamic ice sheet experiment.
To accurately capture the effects of extremes, it is crucial to account for monthly temperature extremes. Our experiments
demonstrated that mass loss is primarily driven by surface melting, which is significantly amplified by the surface elevation
feedback. The impact of extremes, is however not strongly effected by this feedback. Nevertheless, incorporating this feedback
is critical for projections of future sea-level rise. Building upon an SLR estimate derived from a non-dynamic surface mass
balance-only model, our experiments indicate that accounting for the dynamic surface elevation feedback would contribute
roughly 35% to the sea-level rise, and the most severe extreme scenarios would add an additional 20%.

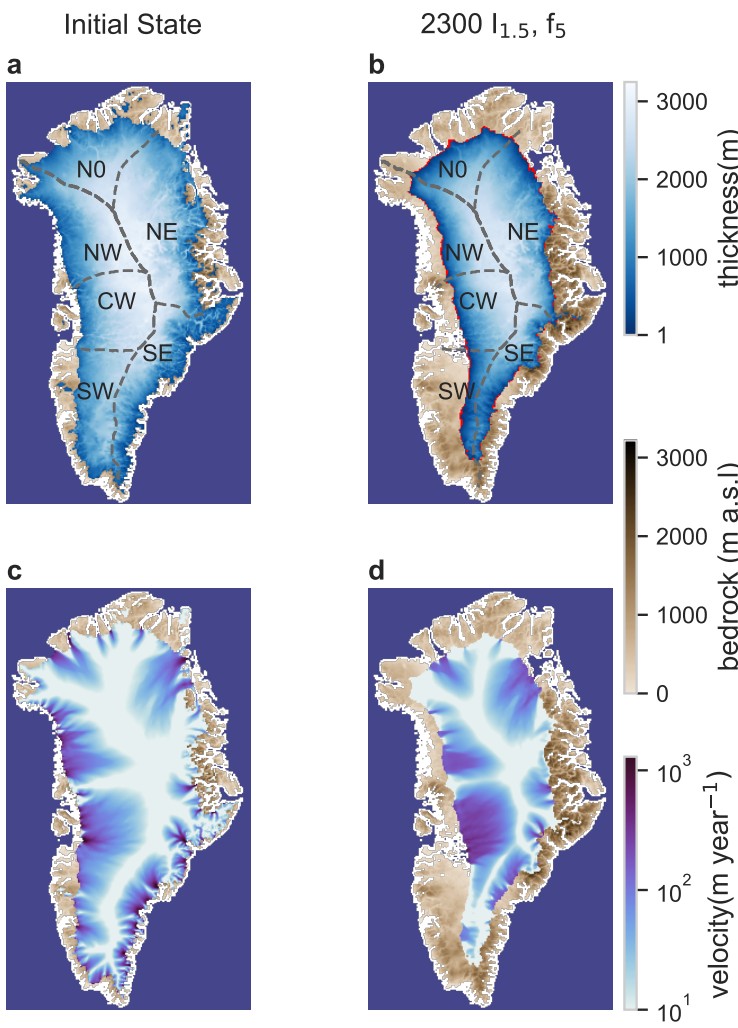

**Figure 1. Additional ice loss caused by extreme events.** (a) Ice thickness (in meters) of initial state with present-day boundary conditions and (c) the corresponding velocity field (in metres per year), as simulated with PISM. Basins are adjusted after (Rignot and Mouginot, 2012) by the IMBIE-2016 team (IMBIE2016, 2019). (b) Projected ice thickness distribution in the year 2300 under MIROC5 RCP8.5 temperature changes, including extreme events ($I_{1.5}, f_5$). Extremes are here applied by increasing the average temperature during the month of July by a factor of 1.5 every 5 years. Red margin indicates the additional area becoming ice-free due to extreme events compared to the MIROC5 RCP8.5 scenario without extremes. Brown shading illustrates the bedrock elevation (in metres above sea-level). (d) Corresponding velocity field in the year 2300 based on the scenario $I_{1.5}, f_5$.

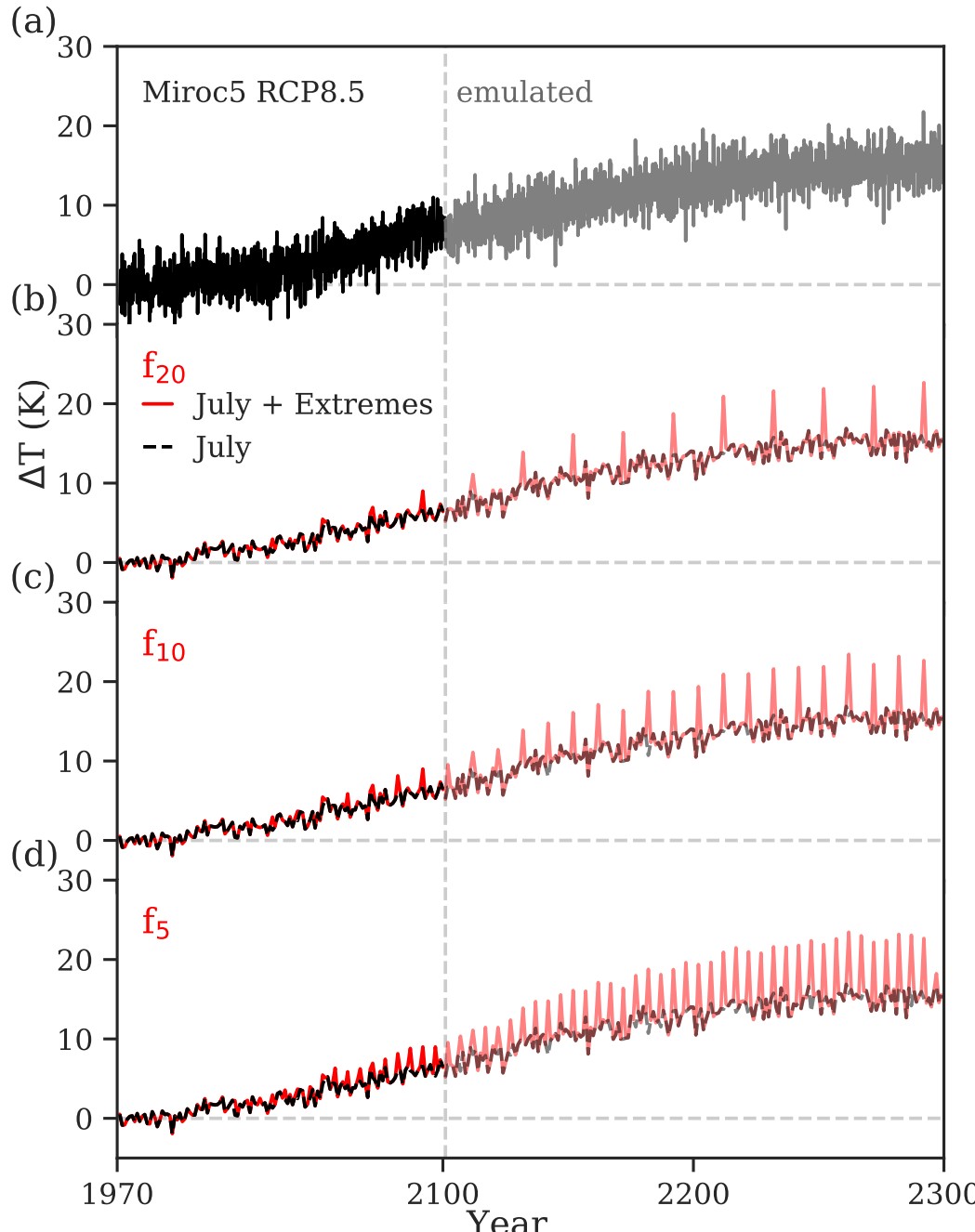

**Figure 2. Temperature scenarios for the Greenland Ice Sheet.** Given is the temperature anomaly over Greenland, based on the MIROC5 RCP8.5 projections, which is applied uniformly at the ice-sheet surface. (a) The forcing scenario without extremes on a monthly timescale (black, solid) from MIROC5 projection until year 2100, and emulated (grey) thereafter (see Methods). (b)-(d) July temperature projection (black, dashed) including extremes (red) occurring every 20 ($f_{20}$), 10 ($f_{10}$) and 5 ($f_5$) years with an intensity of 1.5 times the 10-year running mean ($I_{1.5}$).

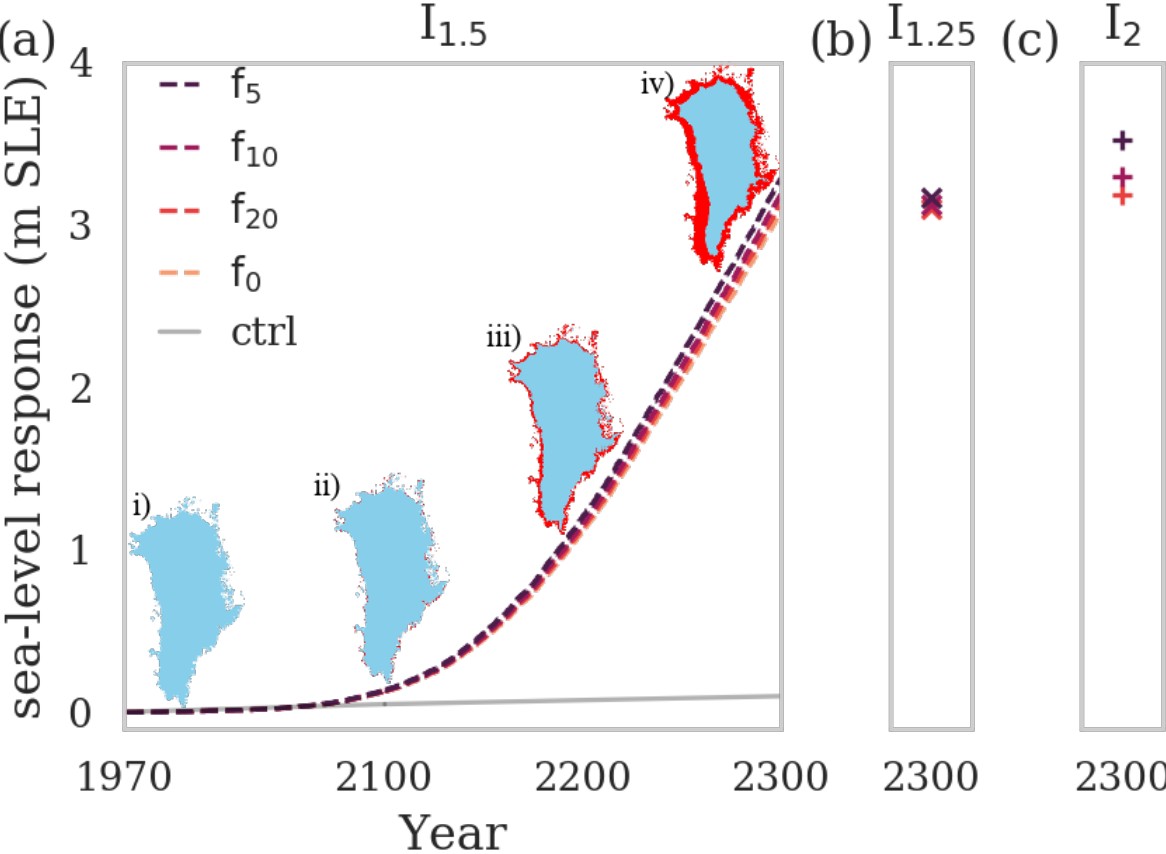

**Figure 3. Sea-level rise contribution of the Greenland Ice Sheet until 2300.** (a) Fully dynamic sea-level rise contribution until 2300 for the forcing scenario without (orange) and with extremes occurring every 20 (red), 10 (pink) and 5 (blue) years, with intensity $I_{1.5}$. Control run (light gray) is subtracted from all simulations. The corresponding ice sheet extent in 1971 (i) and the emerging ice retreat in years 2100 (ii), 2200 (iii) and 2300 (iv) are given in light blue and red shading, respectively. (b),(c) Sea level rise contribution by 2300 for the same experiments but less and more intense extremes of $I_{1.25}$ and $I_5$, respectively.

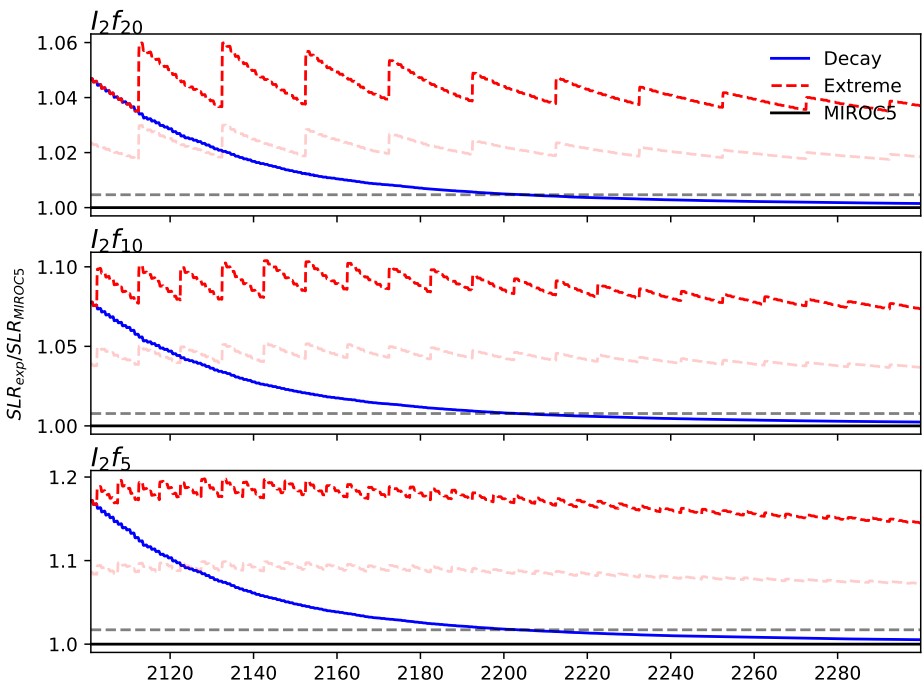

**Figure 4. Relative Sea level rise from 2100 until 2300**. Shown is the relative sea level rise (SLR) in respect to the sea level rise of the MIROC5 experiment form 2100 until 2300. Each panel shows the relative sea level rise for the extreme scenario in red ($I_2$) but for different frequencies ($f_{20}, f_{10}, f_5$). The blue line shows the decay scenario of that extreme scenario, meaning that this experiment was forced with the extreme scenario until 2100 and then continued without extremes until 2300. The black lines show relative sea level rise for the MIROC5 scenario itself and are therefore equal 1 throughout time. The light red line indicates how much relative SLR was, if the difference between the extreme scenario and the baseline scenario (MIROC5) were only half as big. Relative SLR below this light red line is closer to the baseline scenario then to the extreme scenario. The grey dashed line gives the value for which the difference of the extreme and baseline scenario in 2100 is only 10 %.

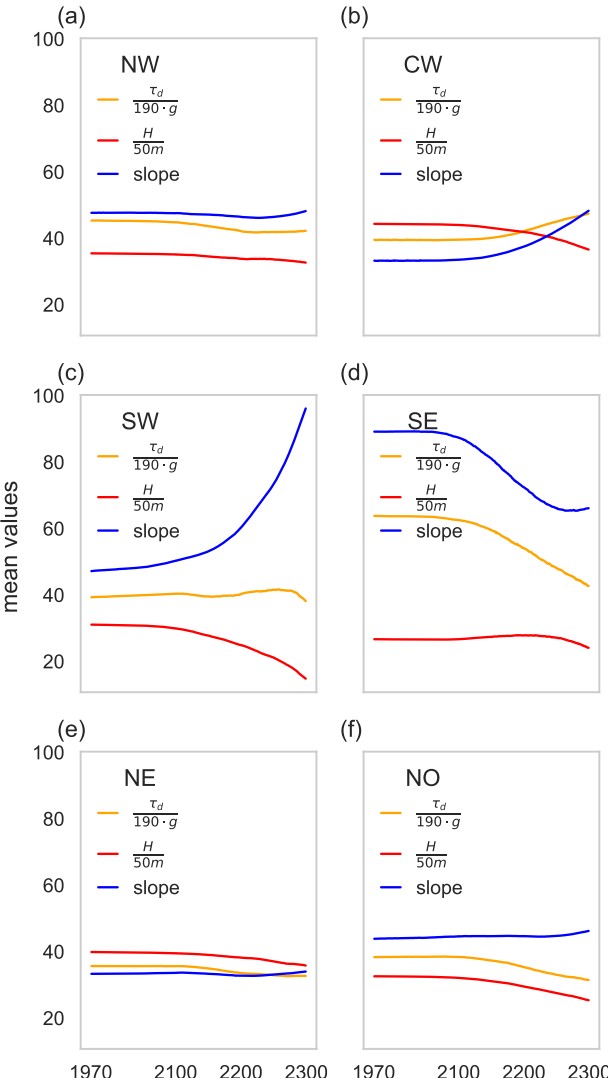

**Figure 5. Mean values of thickness (H), slope and driving stress $\tau_d$ for the $I_{1.5}, f_5$ scenario.** Each panel shows the mean values thickness (H in red), slope in blue and driving stress $\tau_d$ in yellow multiplied by a constant factor of 1/50 ,1 and /190g respectively for each sector. Note that only fields with thicknesses above 1m were taken into account, as this was the original minimum thickness of the BedMachine data Morlighem et al. (2017)

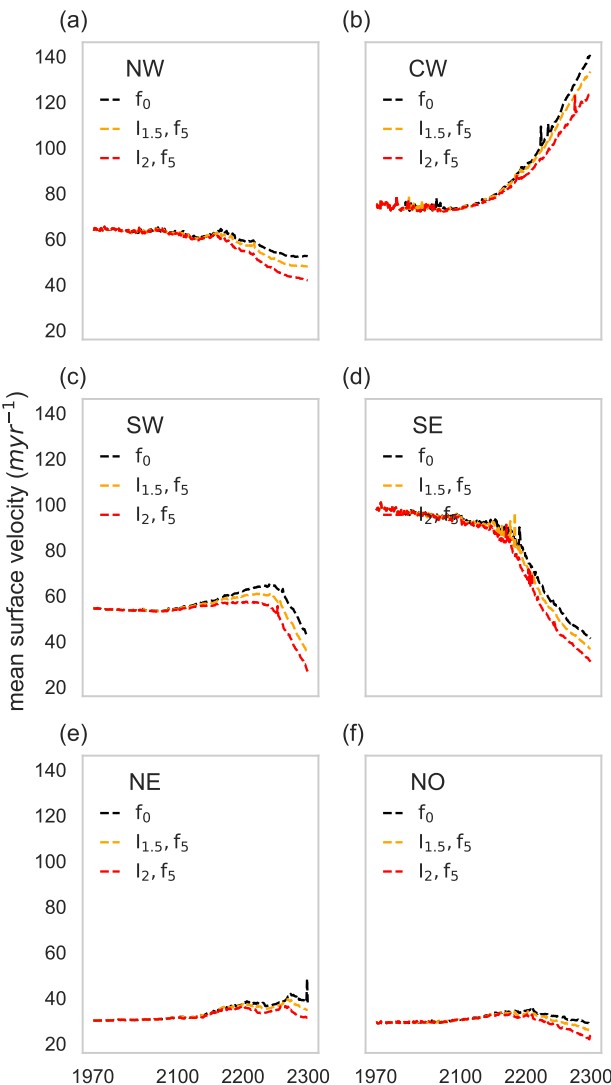

**Figure 6. Mean values of surface velocity for baseline and extreme scenarios** .Each panel shows the mean values of surface velocity for the baseline (black), $I_{1.5}, f_5$ (orange) and the $I_{1.5}, f_5$ (red) scenario for each sector. Note that only fields with thicknesses above 1m were taken into account, as this was the original minimum thickness of the BedMachine data Morlighem et al. (2017)

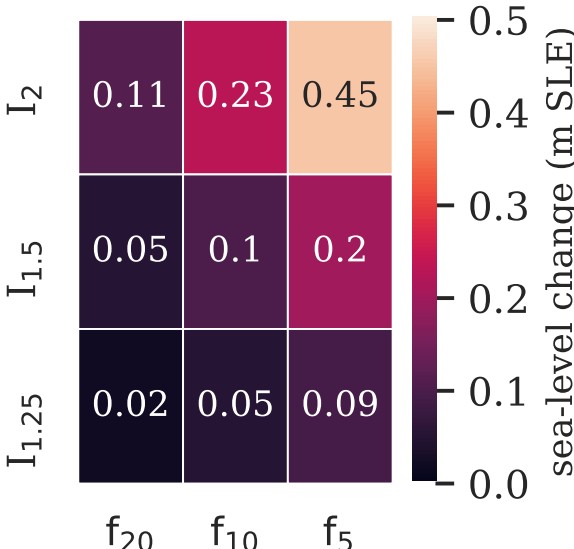

**Figure 7. Importance of intensity and frequency of extremes.** Given is the projected sea-level rise contribution in year 2300, including full ice dynamics, for each extreme scenario, subtracted by the sea-level rise scenario without extremes (MIROC5) in 2300 (see also SI, Table 3).

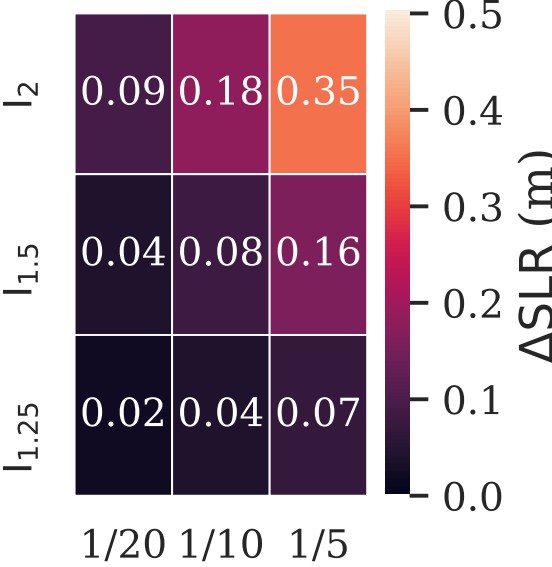

**Figure 8. Importance of intensity and frequency of extremes for the SMB-only case.** Given is the projected sea-level rise contribution in year 2300 for the SMB-only case, for each extreme scenario, subtracted by the sea-level rise scenario without extremes (MIROC5) in 2300 (see also Table 3).

**Tables**

|          |                    | $f_0$ | $f_{20}$ | $f_{10}$ | $f_5$ |
|----------|--------------------|-------|----------|----------|-------|
|          | SMB only           | 0.08m | 0.08m    | 0.08m    | 0.09m |
| $I_{1.25}$ | dynamics, no s-e-f | 0.11m | 0.11m    | 0.11m    | 0.11m |
|          | full dynamics      | 0.13m | 0.13m    | 0.13m    | 0.13m |
|          | SMB only           | 0.08m | 0.08m    | 0.09m    | 0.09m |
| $I_{1.5}$ | dynamics, no s-e-f | 0.11m | 0.11m    | 0.11m    | 0.12m |
|          | full dynamics      | 0.13m | 0.13m    | 0.13m    | 0.13m |
|          | SMB only           | 0.08m | 0.09m    | 0.09m    | 0.10m |
| $I_2$    | dynamics, no s-e-f | 0.11m | 0.11m    | 0.12m    | 0.13m |
|          | full dynamics      | 0.13m | 0.13m    | 0.14m    | 0.15m |

**Table 1.** Projected SLR in the year 2100 for experiments with different extreme event intensities ($I_{1.25}, I_{1.5}, I_2$) and frequencies every 20, 10 or 5 years ($f_{20}, f_{10}, f_5$). Projections without extremes are noted with $f_0$. For each combination of $I$ and $f$, experiments were run allowing changes in SMB only, dynamics without the surface-elevation-feedback (s-e-f) and the full dynamics of the Greenland Ice Sheet.

| $\Delta T$ | $\Delta \mathrm{SMB}_{\mathrm{MIROC5}}$ (Gt yr$^{-1}$) | $\Delta \mathrm{SMB}_{I_2,f_5}$ (Gt yr$^{-1}$) | $\Delta \mathrm{SMB}_{I_2,f_5}/\Delta \mathrm{SMB}_{\mathrm{MIROC5}}$ |
|------|--------|--------|-----|
| 0K   | 0.0    | 0.0    | -   |
| 1.5K | -123.0 | -147.0 | 1.2 |
| 2K   | -165.0 | -199.0 | 1.2 |

**Table 2. Mean GrIS-integrated anomalies of annual SMB** (Gt yr$^{-1}$) **compared to 1980–1999.** Given are the means for the baseline scenario and our most severe extreme scenario($I_2, f_5$) as well as the relative SMB increase between the two.

|  |  | $f_0$ | $f_{20}$ | $f_{10}$ | $f_5$ |
|---|---|---|---|---|---|
| | SMB only | 2.27m | 2.29m | 2.31m | 2.34m |
| $I_{1.25}$ | dynamics, no s-e-f | 2.31m | 2.33m | 2.35m | 2.38m |
| | full dynamics | 3.08m | 3.10m | 3.12m | 3.17m |
| | SMB only | 2.27m | 2.31m | 2.35m | 2.43m |
| $I_{1.5}$ | dynamics, no s-e-f | 2.31m | 2.35m | 2.39m | 2.47m |
| | full dynamics | 3.08m | 3.13m | 3.18m | 3.28m |
| | SMB only | 2.27m | 2.36m | 2.45m | 2.62m |
| $I_2$ | dynamics, no s-e-f | 2.31m | 2.40m | 2.48m | 2.66m |
| | full dynamics | 3.08m | 3.19m | 3.30m | 3.52m |

**Table 3.** Projected SLR in the year 2300 for experiments with different extreme event intensities ($I_{1.25}, I_{1.5}, I_2$) and frequencies every 20,10 or 5 years ($f_{20}, f_{10}, f_5$). Projections without extremes are noted with $f_0$. For each combination of $I$ and $f$, experiments were run allowing changes in SMB only, dynamics without the surface-elevation-feedback (s-e-f) and the full dynamics of the Greenland Ice Sheet.

|  | 2100 | | 2300 | |
|---|---|---|---|---|
|  | $f_0$ | $I_2, f_5$ | $f_0$ | $I_2, f_5$ |
| yearly mean | 0.14 | 0.15 | 3.01 | 3.12 |
| 10 year summer mean | 0.12 | 0.14 | 3.05 | 3.37 |
| summer mean | 0.13 | 0.14 | 3.07 | 3.43 |
| monthly mean | 0.13 | 0.15 | 3.08 | 3.52 |

**Table 4.** Projected SLR of the full dynamic experiment for different resolutions of forcing temperature in the years 2100 and 2300. Experiments were run with the same temperature anomalies of the baseline experiment ($f_0$) and the most severe extreme scenario ($I_2, f_5$) (monthly), with temperature anomaly averaged over a year, over one summer in the months of June, July and August (summer mean) and averaged over 10 summers of the months June, July and August (10 year seasonal mean).

*Code and data availability.* This PISM code used for this study is freely available at https://github.com/pism/pism/releases/tag/v1.1.3 The data from the MAR output is freely available at ftp://ftp.climato.be/fettweis/MARv3.9/ISMIP6/GrIS/ERA_1958-2017 and ftp://ftp.climato.be/fettweis/MARv3.9/ISMIP6/GrIS/MIROC5-rcp85_2006_2100. The BedMachine version3 data set is freely available at https://nsidc.org/data/idbmg4/versions/3 The derived forcing data, spin-up state and scripts for the spin-up, control run, projection runs, and sea level rise calculation for the SMB only case are provided at https://doi.org/10.5281/zenodo.5162937.

*Author contributions.* R.W. conceived the study. J.B. and R.W. designed the study and conducted the analysis. J.B. performed the experiments, data processing, analysis and visualization. J.B. prepared the manuscript with contribution of R.W.

*Competing interests.* The authors declare no competing interests.

*Acknowledgements.* We thank Matthew Palmer for providing his emulated global mean temperature data set, Alison Delhasse and Xavier Fettweis for providing the monthly and daily MAR output fields of the CMIP5 MIROC5 RCP8.5 simulations. This work is supported by the Deutsche Forschungsgemeinschaft project 422877703. R.W. further acknowledges support by the European Union's Horizon 2020 research and innovation programme under grant agreements no. 820575 (TiPACCs) and 869304 (PROTECT). The authors gratefully acknowledge the European Regional Development Fund (ERDF), the German Federal Ministry of Education and Research and the Land Brandenburg
for supporting this project by providing resources on the high performance computer system at the Potsdam Institute for Climate Impact Research.

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
