# Peer review of "Effects of extreme melt events on ice flow and sea level rise of the Greenland Ice Sheet"

_The Cryosphere, 2022_

## Author Comment (AC1)

We thank the reviewers for their constructive critic. We have responded to all *comments* (displayed in *italic*) and adjusted our manuscript accordingly.

In summary we:

- run more simulations and added another section that investigates the importance of resolving extremes.

- We discuss more our dynamics runs without the surface elevation feedback.

-moved some figures from the SI to the main part to improve readability

-identify more clearly the difference of our experiment in comparison to the one of Delhasse et al. and compare to them in the discussion part as they have included extremes as well,

- discuss in more detail our results and add the role of the surface elevation feedback

- rephrase our conclusion and hope to that the discussion and conclusion are readable to a broader audience.

**Reviewer1**

> *Review of "Effects of extreme melt events on ice flow and sea level rise of the Greenland Ice Sheet" by Beckmann and Winkelmann*
>
> *The authors present a set of ice-sheet model simulations to 2300 that explore the impact of extreme events of varying frequency and intensity relative to a simulation with a baseline climate forcing. They construct the baseline temperature forcing using regional climate model estimates of Greenland surface temperature and an emulated global mean temperature time series. They then calibrate a positive degree day model for this temperature time series in order to attain surface mass balance forcing to 2300. They construct nine scenarios for combinations of periods of 5, 10, and 20 years and relative intensities of 1.25, 1.5 and 2, relative to the running decadal mean temperature in the baseline forcing. Running the ice sheet model under these forcing scenarios, they find that including extreme warm events can have a significant impact on long-term mass loss for events with high frequency and intensity. They find a 14% increase relative to the baseline scenario for the most extreme scenario that includes ice dynamics and SMB-elevation feedback. For a case that considers only surface mass balance (i.e., neglecting ice dynamics changes), they find a 16% increase in mass loss.*

*Overall, I think the study is interesting and the paper is well written. I have two primary concerns with the paper, however, which cause me to recommend major revisions.*

*The first concern is that the initial condition is not a good representation of the present day ice sheet, with many major outlet glaciers over- or under-estimating observed velocities by a wide margin. It is not possible to discern from the figures just how far from the modern state the initial condition is, but some major outlets seem to differ from observed velocities by >100% (estimating from Fig S2 by eye). Some misfit statistics are reported in the text, but these are skewed by the very large slow-flowing part of the ice sheet and so the reported average misfit of 9 m/yr is not terribly relevant. It is unclear why this initial condition was used, when there are other PISM initial conditions that look much closer to observations. The present initial condition makes it difficult to interpret the results, as I would not expect this model configuration to respond to external forcing in the same way as a configuration that is closer to observations. I recommend improving the initial condition (possibly just using one that has been published) and repeating these simulations and analysis, or at least somehow demonstrating that the initial condition does not significantly bias the results relative to a more realistic initial condition.*

Our project aims to investigate a first order approximation on the effects of extremes on the overall mass balance of the Greenland Ice Sheet. Therefore, our model calibration was focused on achieving an overall sea level rise potential close to observation rather than simulating each glacier correctly. While glacier speed can be very well represented by ice sheet models, these models usually use inversion as their initialization method, which is not a feature of PISM. The initialization method of PISM is done via a spin-up over a glacier cycle and temperature anomalies of a scalar temperature field. This procedure makes it more challenging to simulate each glacier correctly, especially when using the positive degree day model as the surface model. From this initialization method and surface model,  biases in the ice thickness (Fig. S1,Fig.5c) arise, that in turn influence the surface velocity though differences in thickness and surface slope (driving stress). However, another influence on the velocity patterns of the GrIS stems from the resolution applied in PISM. As the reviewer correctly pointed out, it is possible to simulate glacier velocities close to present-day with PISM, demonstrated by Aschwanden et al 2016. However, there the authors use a flux correction that artificially adds or reduces the ice thickness to match closely the observed surface topology. They also show that the good match to observed velocity is only achieved when using resolution of 600m. Using this resolution would not allow us to run an ensemble of projections until 2300 due to high computational costs. Even when using 600m Aschwanden et al. 2016 show, that especially Humboldt glacier shows the poorest fit in their study and reasons for that would have to be investigated in another study. In their paper and supporting information the authors depict observed and modelled velocity

profiles for different resolutions. For a resolution of 4.5 km, velocity difference in the order of several 100 m/a are common for almost all glacier profiles even with flux correction. Similar published initial states with PISM to ours are published under Zeitz et al, 2021 and 2022 ,We therefore do not think that another spin-up is necessary or could lead to a more realistic velocity representation with this model configuration (glacial spinup, scalar temperature anomalies, pdd surface model ,resolution).

As a second constraint we show that the overall mass balance historical change form 1979-2017 is reasonably well captured with our model simulations (Fig. S3)

> *My second concern is that the experimental design is to essentially add extra temperature forcing, which leads to the conclusion that extreme events are important. But applying these extreme events to the baseline temperature forcing time series results in a stronger average temperature forcing than the baseline. Thus, there is no way to determine how much of the excess mass loss is really due to the extreme events and how much is due to this increase in average temperature forcing. It seems that the proper methodology would be to ensure that the baseline and extreme scenarios have the same mean temperature forcing over some long-term average (probably a few decades to a century). This would much more clearly show the impact of variability vs mean forcing.*

Yes absolutely. We thank the reviewer for pointing this out and added a whole section in the result part to investigate this.

> *The question arises whether the additional temperature excess inserted by the extreme is important to resolve on this monthly time step or whether the evenly distribution of this excess temperature would lead to the same impact in terms of SLR.*

We therefore averaged the original temperature forcing of a monthly resolution over different time frames and asses their contribution to SLR (Tab 4 ). The average temperatures were recalculated to their monthly equivalent to produce a data set with a monthly output. This allows us to keep the same experimental setting of a monthly time step for PISM as in our original experiment and exclude any numerical biases that might be introduced by e.g. changing the time step of PISM. We concentrated on investigating the most severe scenario (I_2,f_5). All results can be found in section 3.5 Resolving extremes

*I also found the Discussion section to be rather limited in scope. I have added a few suggestions below of topics to enhance the Discussion. A number of more specific edits, comments, and questions that are also listed below. There are a number of mis-referenced figures, especially in the very long supplement, so that should be checked carefully during revision.*

Agreed, we added several discussion points and carefully checked the references to the SI.

*Specific comments:*

*L 73: "Consider changes in ocean melt or sliding due subglacial or subglacial processes"*

We explain and discuss the setup more in the methods part. We therefore write now: "We do not consider changes in submarine melt rates or subglacial processes as e.g. glacial channel building that could in turn influence the basal sliding (Methods). As of yet, these processes are not well understood and require different experimental setup (Methods) that would go beyond of the scope of this study (Discussion)."

*L 88: submarine melting is kept constant, but how is it calculated, or what dataset is used? Is the melt rate constant for each glacier for all time, or does each cell have an associated melt rate that is applied when the glacier terminus is in that cell? Are there different treatments for floating and grounded ice? Please provide more information about this.*

*Is there any calving law or criterion used here?*

We thank the reviewer for spotting this lack of clarity. We now add calving front of glaciers are not allowed to advance. We also do not allow for floating ice thinner than 50 m at the calving front and use the von Mise calving law, appropriate for glaciers in Greenland" ... ...."submarine melting is kept constant in time  and space with a melt rate of 0.051914 m/year (PISM default setting).

*L 123–124: By this logic, Humboldt Glacier should have a fairly good match to observations because its width is large compared with the model resolution. However, the fit is very poor there.*

Indeed, Humboldt Glacier has be shown do achieve only poor results when modelled with PISM even for very high resolution of 600m that leads to a good match between observation and simulation for other glaciers as (Aschwanden et al. 2016) demonstrate.

*L 125: There should be similar statistics reported for just the fast-flowing part of the ice-sheet (e.g., where speed > 100 m/yr or some other reasonable threshold), where the velocity and thickness are much more relevant than over the ice-sheet interior.*

Agreed, the Fig. S2 caption now states as RMSE "of 413 m/yr for regions flowing faster than 100m/yr." which is also now mentioned in the text.

*Specify which version of BedMachine is being used. Presumably v3? Citing the paper rather than the dataset (ie., the NSIDC page) is a bit ambiguous because the Morlighem et al (2017) should be cited when using v4 or v5 as well.*

Agreed, we now cite both, the dataset, and the paper, as the title of the paper specifies version 3.

*Fig 1B: Subplot title missing a letter?*

Changed.

*Figure S1: please add a panel showing thickness misfit as a fraction of observed (BedMachine) ice thickness*

Done.

*Figure S2: Color bars on all plots are too narrow, resulting in very large areas of saturated color that make it impossible to judge the fit to the observations. Please use wider color bars; 10^4 m/yr would be a more reasonable upper limit for panels (a) and (b). Also consider using a signed log-scale (e.g., -10^3, -10^2, ..., 10^2, 10^3) for panel c to aid with visualization. There should also be a plot that shows the misfit as a percentage of observed velocity. Some of the velocities at these large outlets (notably Humboldt, NEGIS, most of the NW sector, and potentially others, but hard to tell on this color scale) are very far from the observed velocities, which will significantly bias model results in these regions. This makes interpreting the results rather difficult, as the modeled ice-sheet state is quite far from the true modern state.*

We adjusted the figure and added a panel showing the relative error. We agree that errors are large for some glaciers, but again looking at the best fits with 600m resolution and flux correction in Aschwanden et al. (2016) shows that the mentioned regions by the reviewer are the typical ones that are rather challenging to model.

*L 138–140 and Fig S3: The agreement between the modeled and observed mass balance from 1972–2017 seems overstated to me, given that the slope of the observed mass balance is almost twice the slope of the modeled mass balance from 2000–2017.*

We added ",however not capturing the strong slope from 2000-2017."

*L 157–162: Difficult to understand. I don't understand how the anomaly years contain the monthly anomalies, for instance. Please revise these lines for clarity.*

We rewrote this section and hope it is more clearly now:

To this end, from the annual MIROC5 results until 2100 we first derived a quadratic trend function ($T_{GrIS,trend} = 1280.16°C − 1.31°C/year \cdot years + 20°C/year^{-2} \cdot years^2$) to exclude the inter annual variability (Fig. S4a). Together with the GMT until 2100 we determined a fitting function (Fig. S4b) $T_{GrIS,emulated\ trend} = 0.1°C + 0.96°C^{-1} \cdot GMT + 0.15°C^{-2} \cdot GMT^2$ ,in order to emulate $T_{GrIS,trend}$ beyond the year 2100 in dependence of the GMT. Thus, with the GMT until 2300 and the fitting function, we established the GrIS trend function until 2300 ($T_{GrIS,emulated\ trend}$, red dashed line). To this, we added the inter- and intra- annual variability to receive a more realistic monthly temperature projection. This was done by first calculating the yearly anomalies of the 2050-2100 from the fitting function to the actual annual values: $\Delta T_{yr}(2050−2100) = T_{GrIS}(2050−2100) − T_{GrIS}(2050−2100),trend$ and then randomly picking out of $\Delta T_{yr}(2050−2100)$ and adding on to the emulated trend $T_{GrIS,emulated\ trend}$ until the year 2300. This gave us the new annual temperature curve until the year 2300 with inter-annual variability ($T_{GrIS,emulated}$ red solid line, Fig.S 4a ). However, as we need monthly temperatures we recalculated the monthly temperature values for our newly created $T_{GrIS,emulated}$ by adding the monthly temperature anomalies as well. Thus, for each annual anomaly $\Delta T_{yr}(x)$ we picked, we calculated its monthly anomalies by collecting the full 12 months of that year and subtracting them by the mean of the annual trend (and not the annual mean) ($\Delta T_x(1,...,12) = T_{GrIS}(x(1,..,12)) − T_{GrIS}(x),trend$). Thus each annual anomaly contains now monthly anomalies as well $\Delta T_x = \Delta T_x(1,...,12)$ that were added to the emulated trend. These values served as our baseline scenario until the year 2300 (Fig. 2, dark grey lines).

*L 164: Is Figure S9 the correct figure to reference here? I don't see how it relates to the text here. Seems like it should be Figure S5*

Yes correct, we thank the reviewer for spotting this mistake and changed the Figure number.

*L 181: Should be I1.5f5?*

No, we are here referring to the intensity factor in general.

*Figure S10: There is only one tick on the vertical axis here, which makes it impossible to determine the vertical scale.*

We added a y axis.

*Section S2.1: " Figures S5 and S6 show that the extremes would increase…" These don't look like the correct figures. Should be S8 and S9? Also, the brown curve is not defined in S8 and S9.*

Correct, we changed the Figure number again. There is no brown curve on its own. This brown colour stems from overlaying the black and the yellow curve, which both are defined.

*Figure S12: Why are the two MAR curves here so different over most of the century? I don't fully understand what is meant by: "SLR from the original MAR data set (Miroc5) of 1km resolution was derived from the â SMB", so perhaps that can be phrased more clearly, with a reference to another figure if relevant.*

When calculating the SLR from SMB loss only the mass loss from volumes above floatation must be calculated. Our experiments were run on the 4.5km gird on which we interpolated the MAR SMB original data set that had a resolution of 1km. Thus, for the 4.5km SMB changes from MAR we can estimate the true loss of volume of floatation with our bedrock data set. This was not done for the 1km data set. But to give an estimate of SLR from the original data set we estimated it by calculating the SMB loss over the entire ice volume (also including floating ice cells.)

We write now in the caption:

"SLR from the original MAR data set (Miroc5) of 1km resolution was estimated by the cumulative changes in SMB over the entire ice sheet (also floating cells). The original data set was interpolated to our 4.5km grid as MAR (4.5km). where SLR was calculated by the cumulative changes of  surface mass loss over over the volume above flotation in order to compare it correctly to the PISM PDD simulations."

*L 205: It would be helpful to remind the reader in this sentence of what the scenarios are.*

Agreed, we add now a sentence before:" We derived a set of 10 different temperature forcing scenarios, that include a MIROC5 baseline scenario and its 9 versions of extremes. These extremes differ by three intensities (I1.25, I1.5, I2) with

each having 3 different frequencies (5, 10 and 20
years).

*Figure 2 and in general: It seems strange that only extreme warm events are included in these scenarios, rather than including both extreme cold and extreme warm events. By including only warm events, you've essentially just increased the decadal (or multidecadal to centennial) average temperature by a few degrees, which will of course lead to correspondingly more mass loss. It seems that the proper comparison would be to make temperature time-series that have the same multidecadal average, so that the impact of variability is actually quantified, rather than to add extra temperature forcing to a baseline temperature time series as is done here.*

The AMAP of 2021 showed, that extremes have increased with a bias towards the higher temperatures whereas cold temperature extremes have not increased. We therefore concentrate on looking into increases in temperatures. However, to demonstrate the necessary of resolving these extremes we added a new section that includes runs with equal mean temperatures but disregarding the climate variability. (Section 3.5)

*Figure S13: The vertical axis label should be dST/dz, correct?*

*Correct! We adjusted the axis.*

*L 210: In the SMB-only experiments, does ice thickness change due to SMB? Or is ice thickness held constant in time? Or is advection active, but velocity is held constant? Please add a bit more detail about this set of experiments.*

We added " and thickness was held constant. Thus, SMB changes were calculated with PISM's internal PDD model for a constant surface topography but with changing temperatures."

*Figure S14: Text seems to reference something that isn't present in the figure: "the corresponding ice sheet extent in 1971 (i) and the emerging ice retreat in years 2100 (ii), 2200 (iii) and 2300 (iv) are given in light blue and red shading, respectively."*

True, we deleted that part.

*L ~245: Is this shown in a figure or table anywhere?*

We added a new Figure now in the SI and refer to it in the text.

*L267: typo: Mirco5*

Changed.

*L 300: Could it also be that CW is the only one that continues to accelerate because Jakobshavn remains a marine-terminating outlet, and that's not the case for most other large outlets? From Fig1, it looks like the only other outlets that remain in contact with the ocean are Petermann, maybe NEGIS, and maybe Humboldt.*

We thank the reviewer for this insightful idea. Indeed, most of the other sectors become land terminating and thus velocities decrease. Remaining in contact with the ocean clearly hinders a slowdown for this sector, We added this to the part"

The CW-sector is the only sector in which average surface velocities continue to increase until 2300 (Fig.~S23) with a large part of the glaciers remaining in contact to the ocean, keeping a reduced basal friction and therefore high basal velocities (Fig SX) while other sectors become land terminating.

*What basal friction law is used here? I see that you use an exponent of 0.6, but what is the form of the law? That could have an effect on the slow-down you observe while driving stress decreases.*

We rewrote this passage now as the following:" The basal sliding velocities are related to basal shear stress via a pseudo-plastic power-law with a power of q and the yield
stress. The yield stress in turn follows the Mohr–Coulomb criterion, and is determined by models of till material property (the till friction angle) and by the effective pressure on the saturated till. We linearly altered the friction angle between 5∘ and 40∘between -700m and 700m of bedrock elevation after Aschwanden et al. (2016). The resulting lower friction for lower altitudes and below sea level leads to an additional increase in surface velocities at the ice sheet margins, resulting in an improved match
of flow structure for the glaciers."

*The Discussion section is very short and the Conclusions section reads like it should be in the Discussion. Consider expanding the Discussion and including more of a summary of your findings in the Conclusions. Particularly, the discussion should touch on the impact of the initial ice sheet state on these results, as the spun-up initial condition is quite far from the observed modern ice sheet state (Fig S2). This initial condition should be compared with other model initial conditions for Greenland, at least with the initial condition from Aschwanden et al. (2019).*

We agree with the reviewer here and added a summary of our findings to the conclusion. Now the Conclusion reads:"

Previous studies did not consider extreme melt events when projecting sea-level rise (SLR), and predictions of weather and climate extremes are generally accompanied by high uncertainty (Otto, 2016, 2019). Hence, our idealized experiments offer an initial assessment of how future extreme events may impact the Greenland Ice Sheet, highlighting the significance of incorporating extremes in future SLR projections. It is essential to take into account both the intensity and frequency of extreme events in future projections. Comparatively, our most severe scenario, in contrast to a scenario without additional extremes, could result in an additional SLR of approximately half a meter or 14% within a fully dynamic ice sheet experiment. To accurately capture the effects of extremes, it is crucial to account for monthly temperature extremes. Our experiments demonstrated that mass loss is primarily driven by surface melting, which is significantly amplified by the surface elevation feedback. Incorporating this feedback is critical for precise projections of future sea-level rise. Building upon an SLR estimate derived from a non-dynamic surface mass balance-only model, our experiments indicate that accounting for the dynamic surface elevation feedback would contribute roughly 35% to the sea-level rise, and the most severe extreme scenarios would add an additional 20%.

> *Another topic to touch on in the Discussion is that temperature extremes will in reality increase the flux of meltwater to the bed and thus affect ice dynamics through subglacial hydrology, which is not accounted for in these simulations*

Yes we added this to the discussion:

"The increased surface runoff during the next centuries can clearly influence the subglacial hydrology via the formation of subglacial channels which in turn can influence basal sliding. Our experiments due not consider any changes of this nature, many of these processes are not fully understood and their long term effect is still unclear (Shannon et al., 2013; Tedstone et al., 2015)"

> *Finally, some discussion of the full dynamics runs vs the runs without SMB-elevation feedback would be good.*

*There is no equivalent of Fig 4 given for the dynamic case without SMB-elevation feedback. Overall, it seems like those runs were ignored compared with the SMB-only and full dynamics cases. There should be another subsection analogous to 3.2 in which the full dynamics and no-feedback runs are compared in more detail.*

We wanted the reader to concentrate more on the SMB only and the full dynamic runs because these are the experiments typically run in the community. As the SMB only scenario is close to a setting in which the regional model MAR would simulate SMB loss and common ice sheet models include lapse rate correction when calculating with a pdd model. However, we now add a few more lines in the result part that discuss more the dynamic runs without surface elevation feedback and also added Figure visualizing our analysis.

*We added in the results:*

"The dynamic experiments that do not consider the surface elevation feedback lie in between our SLR projections for the SMB-only and full dynamic experiment but are only slightly higher than the SMB-only runs (about 1 % higher). For this constellation, the dynamic mass loss adds only a little more SLR than the pure SMB loss. This clearly shows the importance of the  surface elevation feedback, i.e. it is important to include the effects of both the surface-mass balance and the ice dynamics in sea-level projections in an interactive manner"

....

"The role of the surface elevation feedback for these simulations can be estimated when comparing the full dynamics runs with the dynamic runs without surface elevation feedback in Figure S25 for the baseline scenario. In the year 2100, thinning is more prominent along the margins and the southwest of the GrIS for the full dynamic runs. The thinned ice cells decrease in surface velocities, but further inland surface velocities speed up due to the steepening gradient. This effect is even further amplified in 2300: Thinning is more and more amplified, reaching into the ice interior of the GrIS leading to further retreat. Steepening the interior leads to speedup while thinning at the margins reduces surface velocities. Dynamic runs without the surface elevation loose only about a quarter of the present day ice sheet area ( 440.3 $10^3$km$^2$ Tab S3). Additional retreat due to extremes is about 4.5,9 and 18 ·103km2 for I1.5 with frequencies of 20,10 and 5 years respectively (Tab S4). Here as well, we see roughly a doubling and halving of the additional mass loss for the other intensities ( I2 and I1.25 respectively). Likewise, the surface elevation feedback shows a bigger effect on ice sheet retreat than the additional extremes"

*I have rated Presentation Quality as "Fair" because there I think the manuscript relies too heavily on the numerous figures in the Supplement, while there are only a few figures in the main paper.*

We moved some Figures from the SI to the main text.

---

## Author Comment (AC2)

We thank the reviewers for their constructive critic. We have responded to all *comments* (displayed in *italic*) and adjusted our manuscript accordingly.

In summary we:

- run more simulations and added another section that investigates the importance of resolving extremes.

- We discuss more our dynamics runs without the surface elevation feedback.

-moved some figures from the SI to the main part to improve readability

-identify more clearly the difference of our experiment in comparison to the one of Delhasse et al. and compare to them in the discussion part as they have included extremes as well,

- discuss in more detail our results and add the role of the surface elevation feedback

- rephrase our conclusion and hope to that the discussion and conclusion are readable to a broader audience.

**Reviewer2**

*The following is a review of, "Effects of extreme melt events on ice flow and sea level rise of the Greenland Ice Sheet" by Beckmann and Winkelmann.*

*This manuscript presents a study of the sensitivity of a Greenland Ice Sheet model to an increased frequency of extreme temperature events in century-scale future projections. The authors design a suite of experiments, in which they the frequency and intensity of events in simulations through the year 2300. To initialize the experiments and spin up the dynamic state of the present-day ice sheet, PISM is run through the last glacial cycle. Then, a number of dynamic and surface mass balance (SMB) parameters are calibrated to best fit total ice sheet mass change over the recent historical period. The authors present global mean sea level contribution results with consideration to SMB forcing only, SMB and dynamics without surface elevation feedback, and a fully dynamic ice response. These experiments illustrate a strong intensification of ice elevation feedbacks after 2100 in response to more frequent high-temperature events, where more*

*frequent and intense events promote increases in interior ice velocities and overall ice sheet mass loss.*

*These types of experiments are an important contribution to the characterization of uncertainties in future projections of Greenland Ice Sheet mass balance, as very few studies have investigated the response of ice sheet dynamics to shifts in natural variability. For this reason, I find that the authors offer valuable insight into the sensitivity of ice sheet dynamics and elevation feedbacks, and into the significance of simulating atmosphere-surface-ice sheet interactions properly. However, I am not convinced that the authors sufficiently introduce, present, and discuss their study, and I think there are a number of improvements needed to better communicate their results to the general cryosphere science community. Therefore, I support publication of this manuscript but after major revisions.*

*I have a number of general concerns, which are listed below:*

- *This study is introduced as a follow-on to Delhasse et al., 2018. While the Delhasse paper is certainly part of the justification for the experiments presented here, it pertains to a much different question, related to atmospheric circulation and blocking patterns that drive extreme melt events on Greenland Ice Sheet. Because the study presented here does not investigate any shifts in the spatial regime of SMB, but only tests increases in continental temperature anomalies, I urge the authors to revise 1) either how the manuscript is introduced and conclusions presented so that they more accurately support experiments designed to study the effects of extreme increases of temperature of the ice sheet or 2) the experiments themselves so that they use forcing in some way derived from the Delhasse et al. paper. The current version of the manuscript is framed as a study of both the quantification of how the SMB changes from increased blocking events alters ice dynamics and of how the frequency and intensity of thermal warming events affects dynamic response. Unfortunately, I am not convinced that it fully investigates either, and the current message seems scattered. See more specific comments below.*

Agreed, we now changed the wording in our introduction to clearly separate our work from the one of Delhasse et al.,2018. We write now:

In a first study Delhasse et al. (2018) assesses the potential influence of an ongoing negative summer NAO under future warming. They simulated the observed circulation pattern from 2000 until 2017, repeatedly until until mid of this century, but with continuous warming at the boundary conditions. Thus they forced the atmosphere to the negative summer NAO and found a potential doubling of SMB loss compared to experiments with the same warming as boundary condition but no negative summer NAO. This approach estimates how such atmospheric conditions lead to generally warmer summers and include of course extreme summers as in e.g. 2012. However, this study does not disentangle the effect of the extremes alone and is limited to SMB changes only, neglecting the dynamic response of the ice sheet.

- *iIf the main goal is to test the effects of extreme melt events on the ice sheet dynamics, and the experimental design is designed to test the effect of variability (as argued in e.g. lines 343-345), then the authors should consider imposing variability in a way that does not perturb the mean SMB forcing. That is, they could compare a run forced with temperature change spread over the entire year and compare that to the response to the change concentrated within one month (July).  These simulations would more pointedly explore the effect of extreme events on ice dynamics (as opposed to testing the response to just adding more overall accumulated warming throughout the simulation).  If the main focus is to instead investigate the ice dynamic response to SMB change resulting from increased warm events that may be missing from future projections, then is no need to design experiments that contain a varying regular frequency of extreme events, but instead the authors could investigate the ice dynamic response to something akin to the magnitude of SMB change that Delhasse et al., 2018 suggest could result from a persistent blocking pattern (i.e. 2000–2016 climate).  Addition of a distinctly stated scientific question with targeted experiments would improve clarity on which are the intended goals of the study.  This may be able to be accomplished with the current simulations, but reorganization and reframing of the text to support the current experimental design would be needed.  Doing so would strengthen the manuscript greatly and make it more accessible to a broader audience.*

We agree with the reviewer and undertook a new set of experiments in which extreme runs are compared to runs that have the same mean temperature but not resolve the variability on a monthly time steps. We introduced a new result section and added the results  (3.5 Resolving extremes) in the abstract and conclusion, as they strengthen the manuscript.

- *The manuscript is written for a specific audience and assumes the reader has an extensive knowledge of ice sheet atmospheric and dynamic modeling. While this is acceptable to a certain extent, I would like to see the authors rework the*

*manuscript so that it can be accessible to a broader audience in the cryosphere science audience. Currently, I think that even ice sheet modeling experts will need to read the manuscript multiple times to really grasp what the results are suggesting. Extension of the discussion to help lead the reader through the implication of the results, especially with regards to pointing out the extensive impact that surface elevation feedback has on the simulations would strengthen the manuscript. Also, because elevation feedback is so important, a description of how the model simulates such feedback should be included in the introduction or supplement in some way. Finally, including some equations that describe the physical response of ice to a change in surface slope (in the methods or supplement) would serve to support an extended discussion and the current description of ice dynamic response within the results.*

Our main purpose of this paper is to give an estimate of future SLR if extremes were to be considered. We think we make that quite clear in the introduction part.

We have rewritten the discussion and result part and hope the result a more clearly now. We also added a part in the methods, ich which we describe were how the melt elevation feedback is simulated.

"PISM can simulate the melt elevation feedback by including a temperature correction for lowering surfaces. For this modelling option, the difference of surface elevation to the initial state is calculated and temperature is corrected with a lapse rate of $6 \circ$C per km. The corrected temperatures then modulate the SMB output via the PDD model accordingly".

- *There are a number of inconsistencies in figure legends and within the text, with regards to how simulations and forcing are references. For instance, please keep acronyms, and especially their capitalization consistent throughout the manuscript and the supplement (e.g. GrIS and MIROC5). Also, especially in the supplement, since there is very little text currently describing the figures, please either expand the captions or add some additional text sections to specifically describe what each figure is, and possible a take home message for each. Finally, within the text, just MIROC5 is often used to describe the forcing, but my understanding is that it is actually MAR-MIROC5. The correct description of the forcing should be used every time, so that it is clear to the reader what product is actually being used.*

We used MAR-MIROC5 to derive our MIROC5 scalar temperature field. We therefore use MIROC5 when we talk about our forcing and continue using it. When comparing out results to MAR, we use MAR-MIROC5.

We checked all Acronyms for consistency.

*Specific comments/questions/suggestions:*

*Abstract: Please mention within the abstract which future scenario is being used for the projections.*

DONE

*Abstract: Since the future SMB is self-imposed by the experimental design (hypothetical changes in intensity and frequency for events), total numbers reported in the abstract in terms of sea level contribution do not have significant meaning to the reader, and are a bit misleading. Perhaps, you could report percent contribution from ice dynamics specifically with respect to its forcing (e.g. percent change in warming)? Or perhaps another diagnostic that is more appropriate and better represents the study results?*

*Abstract: Mentioning that surface elevation feedback is very important would also be appropriate here since it is a major finding of the study.*

We added the relative changes an added ; "Thereby it is crucial to resolve extremes on a high temporal resolution. Overall, the melt-elevation feedback amplifies melting and leads to an additional surface mass loss while the induced thinning reduces the driving stress and decreases surface velocities, dampening ice loss.

*Line 15: sea level rise -> global mean sea level rise*

Done.

*Line 30: Isn't this true only if 1 heatwave can happen per year?*

We don't know what the reviewer means. Probability was determined by calculating the number of heat waves divided the total number of observed years so not only for one year.

*Line 31: Please define for the reader what your criteria is for an extreme melt event.*

We explain in the introduction why all the years were called extreme years. Later in the experimental setup we clearly demonstrate how we define out extreme melt temperatures.

*Lines 45-46: Please reword this sentence. It is a bit awkward and unclear what is meant.*

We write now:

"However, in 2019, the summit melted again. Within three days 97\% of the ice sheet surface was melted."

*Lines 61-63: Hanna et al., 2008 is an older reference. Perhaps this statement could be expanded to include references that pertain to conclusions about the newest CMIP simulations, as well as other natural climate states that modulate melt (i.e. Delhasse et al, 2021)*

Well spotted we wanted to cite Delhasse here and added it now.

*Line 67: To best introduce the reader to the past work on this subject, please expand this statement and summarize the results from Delhasse et al., 2018 here (i.e. their results suggest that current projections neglect changes in extreme events, and so underestimate future reduction in SMB). In addition, please frame how the experiments here relate to the Delhasse study. For instance, Delhasse et al., 2018, specifically look at blocking patterns and therefore the spatial impact they have on Greenland SMB. However, the study presented in this manuscript does not consider spatial patterns in temperature and SMB. Please lead the reader through how the two studies connect, despite such disconnects.*

We now write:

In a first study Delhasse et al. (2018) assesses the potential influence of an ongoing negative summer NAO under future warming. They simulated the observed circulation pattern from 2000 until 2017, repeatedly until until mid of this century, but with continuous warming at the boundary conditions. Thus they forced the atmosphere to the negative summer NAO and found a potential doubling of SMB loss compared to experiments with the same warming as boundary condition but no negative summer NAO. This approach estimates how such atmospheric conditions lead to generally warmer summers and include of course extreme summers as in e.g. 2012. However, this study does not disentangle the effect of the extremes alone and is limited to SMB changes only, neglecting the dynamic response of the ice sheet.

*Line 74: "do not consider changes" -> "do not consider ice response to changes in ocean-induced melt or basal sliding", or something similar that is more descriptive of the processes you refer to here.*

Yes we changed that.

*Line 85: Please describe/cite the methods used for moving the grounding line and the calving front. Is the calving from model independent from the retreat of the model extent on land? Or are the plots of ice retreat (e.g. Fig. 1) just showing where ice thickness is <1m on land area that is above sea level?*

We now added our calving law criterion.

*Line 87: Please note what is the strongly negative SMB defined as. Is this negative SMB prescribed at the points in question? How does that work with the PDD scheme used for the experiments, and does it result in steep slopes and gradients at the edge of the ice sheet?*

Done, it basically means melting the ice away.

*Line 88: Please include how these melting rates are defined, and in the cases where the grounding line retreats and there is new interior floating ice, how are the submarine melting rates set?*

We now mention the constant submarine melt that is applied to all old and new ice cell in contact to ocean.

*Line 96: What is the reasoning behind the very fine vertical resolution? Does the thermal model require this resolution?*

This of advantage when bootstrapping (refining ) the runs in the during the spin-up process and is done equal to Aschwanden et al. 2015.

*Line 97: Does the model treat 1m as ice as still part of the ice sheet, but these areas are then excluded from the analysis (and plotted as non-ice in your plots)? Please specify this in the text for clarity.*

We added, "when visualising our results."

*Line 105: MAR3.9 "forced with" ERA-40 and ERA-Interim. Please note the year span that each reanalysis (40 vs. Interim) product is used.*

Done.

*Lines 111-112: Please specify at what point of the initialization (what year) this is done.*

We added," when switching to the SIA+SSA regime "

*Line 115: Are these the values for the entire spin-up, or are they imposed at a certain time? Please note that in the text.*

Yes, we added this now to the manuscript.

*Line 134: What are the criteria used to determine that these are agreeing well? The trends are very different between the two starting in 2000, which - if I understand correctly - may suggest the model is not really responding dynamically to the shift in climate in 2000. Also, for the SMB comparison in Fig. S3, is part of the discrepancy between PDD and Mouginot SMB because here it is shown with the control subtracted, but emphasis was placed on a SMB match without subtracting the control (i.e. results show in Fig. S12)? Overall, it is unclear to me what the results in Fig. S3 mean, so perhaps you could use some more text to acknowledge some of the mismatch and offer explanation / argument on why that is, and justify why it is acceptable for your experiments.*

All estimations of SLR always calculate the cumulative mass changes, that deviate from a state in balance. Observed SMB changes are calculated by regional Models that can determine melt, accumulation refreezing and surface run off etc. They calculate SMB loss and their cumulative change can be translated into SLR, because they assume the GrIS is in balance, and only changes in SMB would change SLR. These models do not consider dynamic changes. To simulate closely this surface mass loss form MAR with our PDD model, we therefore only consider the SMB changes simulated by the PDD model, therefore also assuming that our model is in balance. These are essentially our SMB only runs but are not subtracted by a model drift. This is done mainly in figure S12 and only for tunning the pdd model. So whenever we compare SMB changes of regional models we compare them to SMB only runs (without subtracting the drift). Figure S12 showed our closest fit for the entire century to MAR-SMB. Figure S3 now shows that if we look at SMB only scenarios (so our pdd model output) compared, to Mouginot the SMB loss is quite off. The SMB estimation of Mouginot was done with Regional Atmospheric Climate

Model v2.3p2 downscaled at 1 km. As a comparison the MAR-original data (1km, resolution) is also depicted and already shows how big the difference between the regional models is. However, the total observed mass balance loss is what in the end would give the true SLR. Because we know our model is not in balance (but the GrIS might gave been), we have to subtract the drift in order to compare SLR correctly. Therefore, whenever we compare our experiments with each other we subtract the model drift, also when we compare it to "observations" that include dynamic changes. We added this explanation to the SI. And added more description in the text as well as in captions of the figures.

*Line 141: Please quantify "slight".*

We rearranged this sentence and eliminated the word "slight".

*Line 159: Is this MIROC5? My understanding is that it is MAR-MIROC5. Please note this accurately throughout the text and the supplement.*

*No, we are talking about our derived scalar temperature field which we defined in the beginning. Also we are modifying in from 2100 with respect to the CMIP5 output so we think that sticking to MIROC5 should be ok.*

*Line 164: Is this Fig S5?*

Yes.

*Line 166: Is this Fig S12?*

Yes, changed.

*Lines 163-166: This paragraph is overall confusing, perhaps because the figures names do not seem to line up with the text. However, I suggest that it be rephrased to better explain what is meant by "the calculated SLR in this case is closer to the original MAR results than when using a 2D temperature field" and how it pertains to the particular Figure being referenced.*

We changed the sentence to: SMB loss calculated by the PDD model seem to agree better with the MAR SMB loss when using such a scalar temperature field.

*Line 172: Please note here that the changes are being made to the forcing only during July.*

Yes added.

*Line 198: In this figure you show the closest fit over the whole ice sheet, but what is the match like spatially after this tuning? Are the gradients in SMB comparable at all? This is a pertinent question since these gradients will be driving the future slope change of the ice sheet and therefore its dynamic response.*

We agree with the reviewer that the future slope can determine the future dynamic response. However, where we want to a general assessment and do not want to investigate regional patterns. For a more correct regional representation a different experimental setup, would be need as e.g. fine resolution and flux correction. We believe that the overall response of the GrIS would still be the same and would experience an overall slowdown but this would need to be assessed in a future study and is mentioned in the discussion now.

*Line 199: Looking at S12, it looks like the best match was 5 degrees, but also with consideration to average temperature (as opposed to 2D)? Please specify. The S12 caption also mentions that the result is time-dependent, but it is unclear what that means. I urge the authors to include much more description and discussion of this method. Currently, it is difficult to follow how the different standard deviation options are derived and then judged. A description of this method is important because results likely depend greatly on the resulting SMB that is derived by the method.*

Yes correct, interestingly the 1d temperature gives an overall better match with the pdd model than the 2d temperature fields. We also tried to change sigma in dependence of the time or temperature, this however did not led to any improvement so we started tuning the refreezing parameter. To not confuse the reader we will remove that line from the figure.

*Lines 202-203: What are these values typically set to in past studies?*

We added the comparison to the default settings and other studies.

*Line 207: Please specify surface temperature (to distinguish from ice temperature).*

Done.

*Line 208: Please do not refer to model output as data. This should be updated throughout the manuscript and in the supplement.*

We changed to "ouput" throughout the manuscript.

*Line 214: Is atmospheric lapse rate the same as the temperature lapse rate above in line 208? If so, please use consistent wording for clarity.*

Yes, it is now "atmospheric temperature lapse rate "for both.

*Line 216: Please describe what the conditions are for the control run and how it is created somewhere in the methods or supplement.*

Yes we added a sentence describing the control run in the method section .

*Line 222: Please quantify slight here, and reference the plot which shows this*

Yes, we do that now.

*Line 230: This conclusion seems a bit obvious, that if you greatly reduce the total SMB, Greenland will lose more mass. As mentioned above, it would be very interesting if you added experiments that allowed you to determine if extreme events themselves, with respect to their frequency and intensity, cause a different response than an equivalent amount of SMB change applied throughout the year.*

Yes, we totally agree and added a section in the result part on resolving extremes.

*Lines 243-248: Please reference figures and tables in this paragraph to help lead the reader through these results.*

Yes, we have created a figure and now reference it to the text.

*Lines 261-264: This is true, you are definitely testing something different, and here, you make the point that your experiments are not imposing SMB changes that are as extreme as Delhasse et al.'s permanent blocking conditions. It is difficult to make the connection on why that comparison is important to the manuscript, because your experiments are so different. Could you expound upon that in the text, and offer some more discussion around why this is an important point with respect to your study and results?*

In our introduction we mention that the only other study that included extremes is the one from Delhasse et al. Therefore, the question arises how our study compares to this one. We think it is helpful to show why there is such a difference, because it is not clear that overall summer temperature variability from 2000-2017 is so different to our MIRCO5 projection. In the discussion we compare our result to other studies and we think the one from Delhasse et al . should not be left out, but in order to do so we have to look at the SMB only scenarios and compare the same mean temperatures increases.

*Lines 271-274: Please make a specific mention of the ice elevation feedbacks here, since it is driving most of your results.*

*Done.*

*Lines 340-346: As mentioned above, it is clear that this study is much different from the Delhasse et al., 2018 study. However, it is not clear what value comparison against their results brings to your manuscript. In addition, the last two sentences of this paragraph are awkward and difficult to interpret. Please rephrase these sentences for clarity and to help the reader understand the value in comparison against the past study.*

The only other study that truly captures extremes as well is the one from Delhasse et al. (2018). As they use a regional atmospheric model we compare our results to the SMB-only runs. Compared to them, our experiments conducted here are rather conservative (at least on the short term) in the sense that the simulated SMB loss only increases by a factor of 1.2 and not 2 as by Delhasse et al. (2018) for comparable average warming. However, this can be attributed to the fact that Delhasse et al. (2018) approach re-simulated the total summer variability of 2000-2017, while we here only increase the variability in July by extremes. Furthermore, it demonstrates that the projected MIROC5 variability is in general lower in the beginning of the century than the ERA observations from 2000-2017. Figure S16 demonstrates the substantial increase of 0.7-0.8 ◦C in summer mean temperatures (on top of the average 1.5 ◦C and 2 ◦C), if we were to add the variability of the entire summers in 2000-2016 as from ERA observations (Dee et al., 2011). Adding the entire summer variability from ERA observations would clearly lead to more SMB loss and bring our results closer to the one of Delhasse et al. (2018). However, this method would not investigate the effect of extremes alone nor would it consider an increase in their variability throughout the century.

Lines 350-352: Please add some text to explain to the reader what the results of these comparisons mean or suggest. What should be the reader's takeaway?

Done.

*Discussion: Elevation feedback should be added to the discussion, beyond within the bounds of limitations, and the authors should add text interpreting the fact that ice dynamics without feedbacks do not really change much between the various experiments or over time.*

Done, we added in the discussing the stronger effect of the surface-elevation-feedback compared to the effect of the extremes.

*Line 365: Can you offer any reasons why your simulations show more sensitivity? Have you tested whether it may be related to the refreeze factor or other tuning? Are there comparable estimates of this feedback for the Aschwanden projections or others that use PISM?*

No, the only other studies where such a comparison was doable are listed in the discussion and are not done with PISM. We do not investigate the reason for that as our main focus is on the impact of extremes, and the surface elevation feedback does not influence the results. But generally with think this might be due to our tuning, as our refreezing parameter Is tuned to the temperature,.Thus an increasing temperature by the surface elevation increases mass loss not only due to more melting but also due to decreased refreezing.

*Lines 370-371: Note here that the tuning was done for total ice sheet-wide SMB.*

Done.

*Lines 377-378: With the current experimental design, I would argue that instead of showing the importance of including extreme (short-term) events, your results show how important it is to capture feedbacks between atmospheric circulation, ice dynamics, ice surface change. Such results suggest that the use of better surface models and a coupled ice-atmosphere setup for projections may be imperative to properly quantifying ice sheet response to future climate. Perhaps*

*this is just a matter of how the text is currently worded, and I misunderstand your meaning here. Please think about rephrasing and expanding the conclusion section to frame the conclusions and explicitly tie them to your results for the readers.*

We agree that in terms of additional SLR the effect of the surface feedback is greater than the one of the extremes. But as we show, we apparently have a high bias for this feedback and if this reduces, the effect of the extremes become more important. However, due to its strong influence on SLR in our experiments we need to address it in our discussion. The long term influence was studied in Zeitz et al. 2022 for different lapse rate factors.

*Figure 1: It is unclear how the ice front changes in the model, and what is being shown in this figure. Are places <1m thickness indicated as bedrock?*

Yes ,anything, below 1m ice thickness is not shown, so either you can see the underlaying bedrock or the ocean. We updated our model description on calving .

*Figure 1: In panel (b), what does the reddish color along the edge of the ice mean?*

We wrote in the caption: Red margin indicates the additional area becoming ice-free due to extreme events compared to the MIROC5 RCP8.5 scenario without extremes.

*Figure 2: This please indicated MAR-MIROC5 instead of Miroc5, if that is indeed the case.*

As mentioned in the text before, as we use a the ice sheet wide average with other modulations for 2300, we want to stick to MIROC5.

*Line 383: This seems to be the same link repeated. Please add a link to the MAR-MIROC5 results here.*

Yes, we updated the link.

*Figure S1: Please saturate the colors on this figure so that it is easier to see smaller changes.*

Done.

*Figure S2, panel (c): This figure looks over-saturated.  It may help to expand the color bar bounds.*

Agreed, we updated the figure now.

*Figure S3: Note that MAR should not be referred to as data.  Also, please add further explanation of what the orange line is.*

Yes, we do that now.

*Figure S5: These differences look to directly correspond to where the dynamic speedups of the simulations are found (i.e. Fig. S22), suggesting that your results may be much different if you were to take 2D fields into consideration.  In the text (line 355), you mention that this treatment could potentially add a bias to the results.  I would argue that your results strongly suggest that the experimental design adds a bias.  Please update your wording to reflect this in the Limitations section of the text.  Do you have any thoughts on the magnitude of the total bias that this introduces?  It is likely related to your highly sensitive elevation feedback in the area.  In this case, it is not clear how meaningful your total number of sea level contribution are. Can you offer further justification for the experimental design (i.e. computational resources or other technical complexities)?*

We agree with the reviewer and now also give SLR increase in relative percent and quantify this bias. However, using a 1D scalar temperature field is a standard procedure when using PISM with the PDD model and there have been many publications with this setup (e.g. Zeitz et al. 2022,Aschwanden et al. 2019). Although this adds biases it also allows us to introduce the extremes in a simplified manner which is our main aim in this paper.

*Figure S8:  Please explicitly define std_extreme.*

Done.

*Figure S11 panel (b): How much does this relationship vary over the ice sheet?  It likely varies seasonally - does this represent only Julys or summers?*

As we use a scalar temperature field we tune this coefficient for the entire GRIS and investigating the regional difference is beyond the scope of this study. Like for the tuning of the precipitation we use the annual mean temperatures and state this now in the figure caption.

*Figure S11 panel (c): Could you offer fit/regression statistics for the line?  What do the number of points represent here (i.e. what is your temporal resolution)?*

As mentioned above, we tuned to the annual values and therefore the temporal resolution is annual. We no added the regression statistics.

> *Figure S12: With total sea level on the y axis, would sea level equivalent be more appropriate of a label than SLR?*

Yes, changed.

> *Figure S15: The caption says 2300, but should be 2100*

Changed.

> *Figure S16: The caption of this figure is a bit misleading, because it sounds like you are comparing against Delhasse et al., 2018.  However, as far as I can tell, you are only plotting your results, and the reader will need to have the knowledge of the other manuscript to make a meaningful comparison.  Is there a way to add Delhasse et al., 2018 results here? This figure is also difficult to digest and needs much more explanation and interpretation in the caption or within the supplement text. For example, please explicitly clarify what each legend symbol means.*

Well not entirely as the purple dashed line is not wat we used but would be something closer to Delhasse et al. However, we hope we clarify this in the caption now.

> *Figure S21: This figure needs a better explanation of what it is showing, or a rewording of the caption for clarity.*

*Done.*

> *Figure S23: Please define what you mean by flux?  Is it ice flux through define gates on the ice sheet?  Is it equivalent to total ice discharge?*

Over the GrIS ice mask, PISM calculates the thickness of each cell the  vertically-integrated horizontal flux. The mean gives and average flux of this aera which is not necessarily the ice discharge. We added this now in the figure.

*Below, I offer some other specific technical suggestions:*

*Line 66: "are" -> "would be"*

Agreed.

*Line 70, 76, 103, 150: GRIS -> GrIS, please be consistent.*

Done

*Line 131: "These are" -> "This is"*

*Agreed, done.*

*Line 381: pism->PISM*

Done.

*Fig. 2, S4, S6, S7, S12, S26, Table S2: Miroc5->MIROC5*

Done.

*References:*

Delhasse, A., Fettweis, X., Kittel, C., Amory, C., and Agosta, C.: Brief communication: Impact of the recent atmospheric circulation change in summer on the future surface mass balance of the Greenland Ice Sheet, Cryosphere, 12, 3409–3418, https://doi.org/10.5194/tc-12-3409-

2018, 2018.

Delhasse, A, Hanna, E, Kittel, C, Fettweis, X. Brief communication: CMIP6 does not suggest any atmospheric blocking increase in summer over Greenland by 2100. Int J Climatol. 2021; 41: 2589– 2596. https://doi.org/10.1002/joc.6977

Zeitz, M. and Haacker, J. M. and Donges, J. F. and Albrecht, T. and Winkelmann, R.: Dynamic regimes of the Greenland Ice Sheet emerging from interacting melt--elevation and glacial isostatic adjustment feedbacks. Earth System Dynamics. DOI: 10.5194/esd-13-1077-2022

Zeitz, M. and Reese, R. and Beckmann, J. and Krebs-Kanzow, U. and Winkelmann, R.: Impact of the melt--albedo feedback on the future evolution of the Greenland Ice Sheet with PISM-dEBM-simple, The Cryosphere. 2021,doi: 10.5194/tc-15-5739-2021}

Aschwanden, A., Fahnestock, M. A., Truffer, M., Brinkerhoff, D. J., Hock, R., Khroulev, C., Mottram, R., and Khan, S. A.: Contribution of
the Greenland Ice Sheet to sea level over the next millennium, Science Advances, 5, eaav9396, https://doi.org/10.1126/sciadv.aav9396,
2019.

---

## Author Response (AR2)

Dear Dr. MacGregor,
We thank you for the kind words. We have addressed all your comments and responded to them point by point. Please see our responses below.

11: "Thereby"
12: Specify the minimum temporal resolution needed. From my interpretation of the MS it is monthly, as seasonal (3 months) appears inadequate.
13: Can "longer timescales" be specified as "annual"?

We have changed all points mentioned in line 11,12,13 and write now:

In conclusion, projecting the future sea-level contribution from the Greenland Ice Sheet requires considering both the changes in the frequency and intensity of extreme events. It is crucial to individually address these extremes on a monthly resolution as temperature forcing with the same excess temperature but evenly distributed over longer timescales (e.g. seasonal) lead to less sea level rise than for the simulations of the resolved extremes.

60: I understand the rationale for the use of the term "full dynamics" here, but I caution that it is not a great use of the term as for a casual reader who miss this definition, it may imply a full-Stokes treatment or perhaps more explicit consideration of ocean forcing. If the authors could identify an alternative similarly brief term that does not use "full", I recommend doing so. Not essential.

We understand the editors concern but could not think of another short name for the experiment. Changing the name would probably mean, also to change the name of experiment 3. However, to avoid confusion, we now mention in the experimental description that full dynamics is SIA +SSA.

61: Identify RCP acronym here.
Done.

101: Given that it is the PISM-default and bizarrely precise value, especially considering that it is explicitly stated later on how little is known about ocean forcing, "~0.05 m/yr" seems more appropriate here.

Agreed, this was a copy and paste error from the source code. We changed the value now.

259: Here and throughout the MS (e.g., 324-330), add \times between values and 10^X multipliers where needed.

Done.

Figure 5: I really like the concept of this figure, but consider using a relative vertical scale (perhaps 50–200%) instead, so that the convoluted unit multipliers and the differences between these quantities are easier to understand. Then regional mean values can simply be stated in the legend instead of the multiplier. Move region labels to next to letters, e.g., "(a) NW".

Yes, that is a better idea. We now show relative changes and state the initial values in in the panel.

Combine Figures 7 and 8 and label their difference above them. Also, consider reversing the color scale.

We combined the graphs but kept the color scale.

Tables 1 and 3: Remove "m" from all table values and specify unit in table caption instead.

Done.

Table 4: Specify units (meters?) for values given in table.

Done.